# Can Transformers Solve Least Squares to High Precision?

**Jerry Liu** [1]  **Jessica Grogan** [2]  **Owen Dugan** [3][4]  **Simran Arora** [1]  **Atri Rudra** [2]  **Christopher Re** [1]

## Abstract

Deep sequence models like Transformers have achieved remarkable results across language and vision tasks, but their ability to solve high-precision numerical problems, crucial in scientific settings, remains unclear. We explore the capabilities of existing models on the fundamental problem of least squares, motivated by recent work suggesting Transformers can implement learning algorithms on in-context linear regression problems. Surprisingly, we observe that Transformers struggle to solve least squares to high precision, even in fully determined settings: their MSE plateaus at $10^{-5}$, 9 orders of magnitude worse than simple algorithms like gradient descent. Probing for sources of low precision, we train on basic linear algebra operations and find that Transformers struggle to precisely learn a simple element-wise multiplication task. Since numerical methods rely heavily on linear algebra primitives, including multiplication, this result suggests that Transformers struggle to implement learning algorithms to high precision, in contrast to prior findings. Our key insight is that gated convolutional models can exactly implement arithmetic circuits, including multiplications and polynomials. Using gated convolutions, we instantiate a weight construction that directly solves least squares to high precision by explicitly implementing gradient descent. Finally, based on our analysis, we propose a simple alternative to standard in-context learning, in which we supervise models to explicitly learn the gradient update rule and apply them iteratively during inference. Using this framework, we achieve 2 orders of magnitude improvement over parameter-matched Transformers trained on standard in-context learning.

---

[1]Stanford University [2]University at Buffalo [3]MIT [4]NSF AI Institute for Artificial Intelligence and Fundamental Interactions. Correspondence to: Jerry Liu <jwl50@stanford.edu>.

*Proceedings of the $1^{st}$ Workshop on In-Context Learning at the $41^{st}$ International Conference on Machine Learning*, Vienna, Austria. 2024. Copyright 2024 by the author(s).

## 1. Introduction

Deep sequence models, especially the prevailing Transformer architecture, have demonstrated a remarkable capacity for generalization and robustness across language and vision tasks (Touvron et al., 2023; Chowdhery et al., 2022; Brown et al., 2020). Transferring these benefits to scientific domains is an exciting prospect that has the potential to unlock new fundamental capabilities across science and engineering (McCabe et al., 2023; Subramanian et al., 2023; Yang et al., 2023). Crucially, applications such as fluids and climate modeling require high precision solutions (Frisch, 1995), and it is not clear whether existing ML methods can achieve the accuracy of standard numerical methods.

Towards obtaining high-precision solutions with ML, we focus on the testbed of least squares. A large class of differential equations problems can be reduced to least squares problems (Gottlieb & Orszag, 1977; Orszag, 1972; Trefethen, 2000), so it seems crucial for models to be able to solve them precisely before they can hope to solve broader problems like PDEs. Furthermore, we are motivated by a surge of recent work (Garg et al., 2022; Akyürek et al., 2022; Fu et al., 2023; Ahn et al., 2024; Bai et al., 2024) that suggest that Transformers can perform optimization algorithms *in-context*.

Following prior work (Garg et al., 2022; Von Oswald et al., 2023), we train Transformers on in-context linear regression problems, investigating how precision scales with depth. Surprisingly, we find that they struggle to achieve below $O(10^{-5})$ MSE, even on the simple case of *noiseless fully-determined systems* (Section 3, Figure 1). This accuracy is remarkably poor compared to the machine precision solutions ($10^{-14}$ MSE with single-precision) gradient descent consistently obtains in this setting (Boyd & Vandenberghe, 2004).

In this work, we make progress towards understanding why Transformers struggle with learning high-precision algorithms.

**Identifying expressivity limitations within Transformers.** First, towards identifying mathematical operations that represent precision bottlenecks for the Transformer architecture, we examine gradient descent and Newton's method, two classical numerical algorithms that are known

Least squares, Transformer vs. OLS

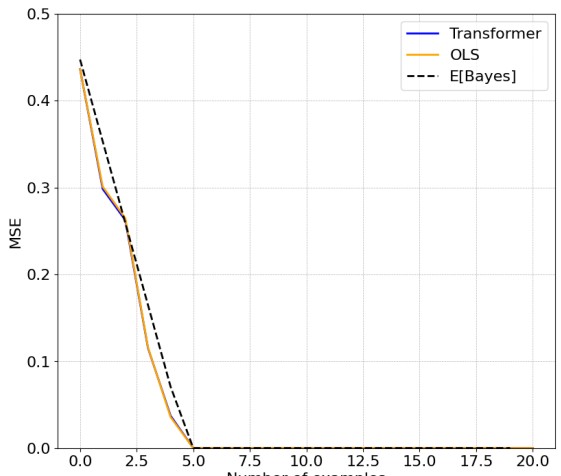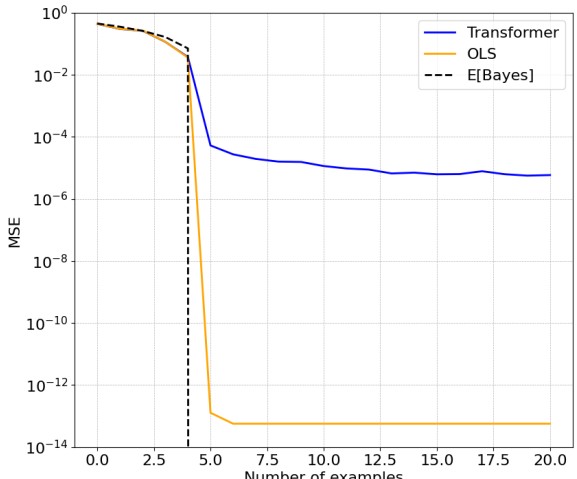

*Figure 1.* Although Transformers and OLS seem to have comparable performance on *underdetermined* noiseless linear regression (left), precision of Transformer solutions saturate 9 orders of magnitude above OLS in the *fully-determined* regime (right).

to reliably reach machine precision on least squares (Schulz, 1933; Weisberg, 2005). We observe that these optimization algorithms can be written as a composition of three basic primitives: arbitrary *read/writes*, *affine transformations*, and *element-wise multiplication*. Training Transformers on synthetic formulations of these tasks, we identify high-precision multiplications as a challenge for attention-based models: precision on the multiplication task scales surprisingly poorly with increased depth, number of parameters, and training duration. Since numerical methods rely heavily on high-precision linear algebra primitives, this result suggests that Transformers would face a fundamental expressivity challenge if trying to directly implement optimization algorithms like gradient descent.

**Closing the gap with gated convolutions.** We next investigate alternatives to softmax attention that may help improve the precision of our models on least squares. We focus on gated convolutions, another popular class of sequence models (Arora et al., 2023; 2024) combining element-wise multiplications with convolutional filters. Recent work has shown the ability of gated convolutions to efficiently represent *arithmetic circuits*, including multiplications and polynomials (Arora et al., 2023). Using theory and empirical constructions, we demonstrate that gated convolutions are expressive enough to solve least squares to high precision by explicitly implementing gradient descent arithmetic circuits. For linear regression, we empirically implement a weight construction for gradient descent using gated convolutions and demonstrate that it is expressive enough to solve least squares to $10^{-14}$ MSE, a lift of 9 orders of magnitude from in-context Transformers.

**Learning high-precision algorithms.** Finally, we investigate whether gated convolutions can *learn* the high-precision numerical algorithms we've shown they can implement theoretically. Surprisingly, we observe that despite our insights about the expressivity of BASECONV, they perform 2 orders of magnitude *worse* than Transformers when trained naively on in-context least squares (Figure 10).

To tease apart the complexity of learning algorithmic solutions, we propose a simple approach in which we supervise models to explicitly learn the gradient update rule. During inference, we then iteratively apply the learned circuit on least squares problems until convergence. Using this simple training setup, we observe an improvement of 2 orders of magnitude over parameter-matched Transformers trained via standard in-context learning.

## 2. Background

In this section, we provide background information about our model architectures, problem framework, and training setup. For a detailed discussion of related work, please refer to Appendix A.

### 2.1. Sequence model architectures

Inspired by language modeling, we study autoregressive sequence-to-sequence models $T_\theta : \mathbb{R}^{N \times D_{in}} \to \mathbb{R}^{N \times D_{out}}$, where the sequence length is $N$, each element of the input sequence lies in $\mathbb{R}^{D_{in}}$, and each element of the output sequence lies in $\mathbb{R}^{D_{out}}$. Sequence models like Transformers (Vaswani et al., 2017) share a common high-level structure. First, a linear projection $P_{in} : \mathbb{R}^{D_{in}} \to \mathbb{R}^D$ embeds each input element to a shared $D$-dimensional embedding

space, yielding a matrix $\boldsymbol{u} \in \mathbb{R}^{N \times D}$. Next, $\boldsymbol{u}$ is passed through a stack of $L$ layers $T^{(\ell)} : \mathbb{R}^{N \times D} \to \mathbb{R}^{N \times D}$. Each layer consists of a *sequence mixer*, which mixes information across the sequence dimension, and a *state mixer*, which mixes information across the model dimension. Finally, a linear projection $\boldsymbol{P}_{out} : \mathbb{R}^D \to \mathbb{R}^{D_{out}}$ maps to the output space.

In this work, we focus on two classes of sequence mixers: attention and gated convolutions.

**Attention.** Multi-headed softmax attention is the sequence mixer used within the prototypical Transformer architecture (Vaswani et al., 2017), which remains dominant across language and vision tasks. Each *head* of an attention layer is parameterized by three projection matrices $\boldsymbol{W_Q}, \boldsymbol{W_K}, \boldsymbol{W_V} \in \mathbb{R}^{D \times D}$. For an input $\boldsymbol{u} \in \mathbb{R}^{N \times D}$, the attention operator $\text{ATTN}(\boldsymbol{u})$ is defined as:

$$\sum_{i=1}^{H} \text{softmax}\left( \left(\boldsymbol{u}\boldsymbol{W_Q}^{(i)}\right) \left(\boldsymbol{u}\boldsymbol{W_K}^{(i)}\right)^T \right) \left(\boldsymbol{u}\boldsymbol{W_V}^{(i)}\right) \quad (1)$$

where $H$ is the number of heads per layer.

**Gated convolutions.** Gated convolutions combine element-wise multiplications (gating) with long convolutions, where the convolutional filters are of the size of the sequence length. In this work, we focus on a variant of the BASECONV operator from (Arora et al., 2023). Given an input $\boldsymbol{u} \in \mathbb{R}^{N \times D}$, BASECONV$(\boldsymbol{u})$ is defined as:

$$\underbrace{(\boldsymbol{u}\boldsymbol{W}_{gate} + \boldsymbol{b}_{gate})}_{\text{Linear Projection}} \odot \underbrace{(\boldsymbol{h} * (\boldsymbol{u}\boldsymbol{W}_{in} + \boldsymbol{b}_{in}) + \boldsymbol{b}_{conv})}_{\text{Convolution}})\boldsymbol{W}_{out} + \boldsymbol{b}_{out}$$
$$(2)$$

where the layer is parameterized by learnable filters $\boldsymbol{h} \in \mathbb{R}^{N \times D}$, linear projections $\boldsymbol{W}_{in}, \boldsymbol{W}_{gate}, \boldsymbol{W}_{out} \in \mathbb{R}^{D \times D}$, and bias matrices $\boldsymbol{b}_{conv}, \boldsymbol{b}_{in}, \boldsymbol{b}_{gate}, \boldsymbol{b}_{out} \in \mathbb{R}^{N \times D}$. The $\odot$ is component-wise product and convolution of two matrices is computed as convolution of the corresponding columns.

### 2.2. Least squares and in-context learning

Recent works have investigated the ability of Transformers to solve least squares problems within an *in-context learning* framework (Garg et al., 2022; Akyürek et al., 2022; Bai et al., 2024). We briefly describe the in-context training setup from (Garg et al., 2022).

We consider the following parameter estimation problem: given samples $\{(\boldsymbol{x}_i, y_i := f(\boldsymbol{x}_i; \boldsymbol{w}^*))\}_{i=1}^{N}$ with given function $f$ and unknown parameter $\boldsymbol{w}^*$, our goal is to predict $y_q = f(\boldsymbol{x}_q; \boldsymbol{w}^*)$ given query point $\boldsymbol{x}_q$. For linear regression, $f(\boldsymbol{x}_i; \boldsymbol{w}^*) = \boldsymbol{x}_i^T \boldsymbol{w}^*$. Following prior work (Garg et al., 2022; Akyürek et al., 2022), we define a distribution

of *prompts*

$$P = (\boldsymbol{x}_1, y_1, \ldots, \boldsymbol{x}_N, y_N) \quad (3)$$

where the $\boldsymbol{x}_i$'s and $\boldsymbol{w}^*$'s are sampled from some joint training distribution $\mathcal{D}_{train}$. We supervise a large sequence model $T_\theta$ to predict the output $y_q = f(\boldsymbol{x}_q; \boldsymbol{w}^*)$. The training objective is to minimize the expected mean squared error, averaged over each of the $n$ independent least squares problems per prompt:

$$\min \ \mathbb{E}_P \left[ \frac{1}{N} \sum_{k=0}^{N-1} \left|\left| T_\theta(P^k) - y_{k+1} \right|\right|^2 \right] \quad (4)$$

where $P^k = (\boldsymbol{x}_1, y_1, \ldots, \boldsymbol{x}_k, y_k, \boldsymbol{x}_{k+1})$.

Excitingly, a recent line of work probes the estimators learned by Transformers on in-context least squares, and suggests that Transformers learn to mimic iterations of learning algorithms, like gradient descent and Newton's method (Von Oswald et al., 2023; Ahn et al., 2024; Fu et al., 2023; Giannou et al., 2024). We refer to Appendix A for a more detailed discussion of related work.

## 3. Identifying precision as a challenge for Transformers

In this section, we empirically investigate the claim that Transformers learn to implement algorithms in-context to solve linear regression. Crucially, we show that Transformers struggle to obtain high precision solutions, even on noiseless, fully determined problems (Section 3.1). In contrast, we know that numerical algorithms for linear regression like gradient descent robustly converge to machine precision solutions. To investigate the precision issue on a simplified setting, we isolate a set of linear algebra operations, which naturally appear as primitives comprising a general class of numerical algorithms (Section 3.2). These synthetic tasks motivate the alternative architectures we analyze theoretically (Section 4) and are a natural testbed for evaluating high-precision training (Section 5).

### 3.1. Transformers struggle to precisely solve linear regression

**Experimental setup.** Following prior work (Garg et al., 2022), we train a 12-layer Transformer on *noiseless* linear regression problems with $D = 5$, where the $\boldsymbol{x}_i$'s and $\boldsymbol{w}^*$'s are drawn from a standard multivariate Gaussian. We vary $N \in \{0, \ldots, 20\}$ and evaluate the MSE of the model's learned estimator. Please refer to Appendix B.2.1 for more details about the training setup.

In Figure 1, we compare the performance of the Transformer models to the Bayes-optimal estimator, Ordinary Least Squares (OLS). We note that the precision gap between the Transformer and the Bayes-optimal estimator

drastically increases when $N \geq D$:

- For *underdetermined* regression problems, i.e. when $N < D$, there exists an entire hyperplane of possible $\boldsymbol{w}$'s that perfectly match the provided data, so the optimal estimator will have non-zero MSE. In this setting, our results match the observations from prior works (Garg et al., 2022; Akyürek et al., 2022), which note that Transformers seem to approximate Bayes-optimal estimators (i.e. OLS for dense $\boldsymbol{w}$'s.)

- For *fully determined* regression problems, i.e. when $N \geq D$, there exists a unique $\boldsymbol{w^*}$ that solves the problem. In theory, the OLS estimator recovers $\boldsymbol{w^*}$ exactly and thus should have an MSE of 0. In practice, we observe that when computed using floating-point arithmetic, OLS accrues some numerical error on the order of machine epsilon: for single-precision, $O(10^{-14})$. In contrast, the Transformer struggles to reach below $10^{-5}$: this is a difference of 9 orders of magnitude.

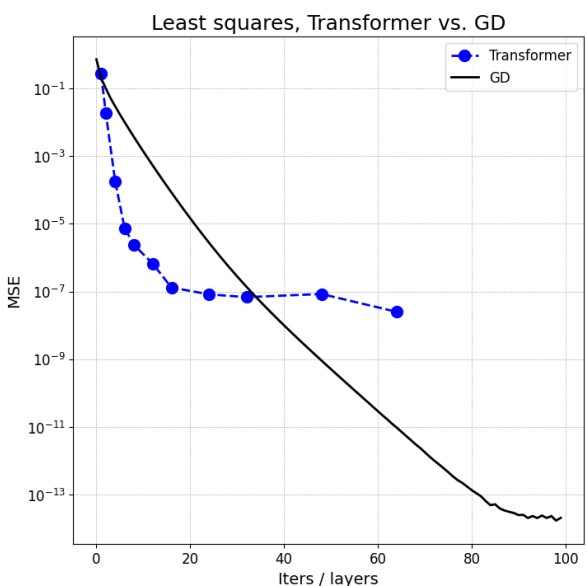

*Figure 2.* While precision saturates for Transformers trained on (*fixed N*) fully-determined least squares ($O(10^{-7})$), gradient descent converges to machine precision ($O(10^{-14})$): this is a difference of 7 orders of magnitude.

**Scaling studies.** We thus focus on the *fully determined* case, and investigate whether precision improves with larger models. Note that if Transformers are able to implement iterative algorithms to solve linear regression, the depth of the model should correspond to the number of iterations of the algorithm. Algorithms like gradient descent converge to the exact solution with enough iterations: do Transformers have the same property?

In Figure 2, we consider a simplified training setting with *fixed $N > D$*, where the model is evaluated only on the final prediction $y_N$. We train Transformers on this task, up to $L = 64$ layers, and we compare their precision scaling to the convergence rate of full-batch gradient descent. For more details about the training setup, refer to Appendix B.2.2.

We observe that Transformer precision scaling exceeds the convergence rate of gradient descent at first, but the precision gains for Transformers rapidly diminish, such that we observe very little difference in precision between $L = 32$ and $L = 64$ layers. For our deepest Transformer models, we achieve an MSE around $O(10^{-7})$. In contrast, gradient descent converges linearly to machine precision, about 7 orders of magnitude more precise.

The diminishing returns of the Transformer precision scaling law imply that the story of in-context learning as gradient descent is incomplete. These results indicate there exists a large gap between the high-precision algorithms Transformers can theoretically express (Bai et al., 2024) and their empirical performance when trained in-context.

### 3.2. Synthetic: investigating primitives from numerical methods

To better understand the precision issue with Transformers, we start by looking into primitives that comprise optimization algorithms such as gradient descent and Newton's method. Since these algorithms are so fundamental to the field of numerics, they represent a natural starting place for discovering simple operations that Transformers struggle to precisely express.

We observe that these algorithms can be expressed as compositions of three simple linear algebra operations mapping from inputs $\mathbf{u} \in \mathbb{R}^{N \times d_{in}}$ to outputs $\mathbf{y} \in \mathbb{R}^{N \times d_{out}}$:

- Sequence-wise read/write:

$$\text{READ}(i, j, a, b)(\mathbf{u}) = \begin{cases} \mathbf{u}[k, a:b] & k \neq j \\ \mathbf{u}[i, a:b] & k = j \end{cases} \quad (5)$$

  where $d_{in} = d_{out}$.

- Affine transformations:

$$\text{AFFINE}(\boldsymbol{H})(\mathbf{u}) = \mathbf{u}\boldsymbol{H} \quad (6)$$

  where $\boldsymbol{H} : \mathbb{R}^{d_{in}} \to \mathbb{R}^{d_{out}}$ is an affine linear map.

- Element-wise multiplications:

$$\text{MULTIPLY}(a, b, d_{out})(\mathbf{u}) = \mathbf{u}[:, a:a+d_{out}] \odot \mathbf{u}[:, b:b+d_{out}] \quad (7)$$

In Appendix D.2, we describe how gradient descent and Newton's method iterates can be expressed as a composition of these primitives. Intuitively, READ is used to transfer

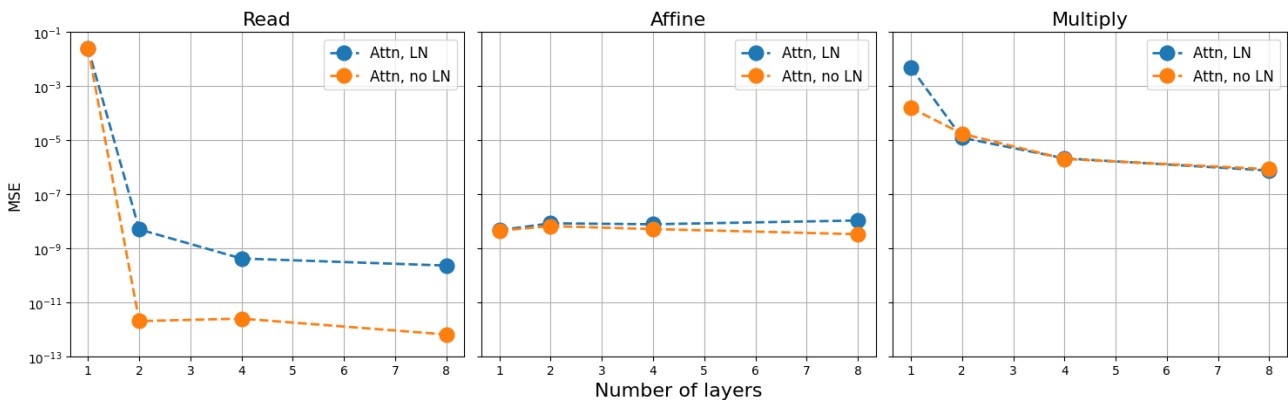

*Figure 3.* Precision vs. Transformer depth, with and without LayerNorm (LN), on synthetic tasks. While shallow Transformers are able to learn the READ and AFFINE tasks to high precision ($< 10^{-8}$ with 2-layer models), precision on the MULTIPLY task scales poorly with depth ($O(10^{-6})$) with 8-layer models).

information across the sequence dimension, AFFINE to transfer information across the hidden dimension, and MULTIPLY to compute high-degree interaction terms (like dot products or element-wise squaring).

**Empirical analysis.** We train Transformers on these synthetic tasks to investigate how precision scales with model size. Details about our training setup are in Appendix B.2.3.

In Figure 3, we show that even 2-layer Transformers are able to achieve high precision ($O(10^{-8})$ MSE) on the READ and AFFINE tasks. However, we find that Transformers struggle with the MULTIPLY task. In Figure 4, we show that precision scales surprisingly poorly with model size.

**Theoretical analysis.** Towards understanding the precision limitations of Transformers on the MULTIPLY primitive, in Appendix D.2.4, we provide a proof that single-layer linear attention is unable to exactly represent the simple element-wise squaring function

$$\text{SQUARE}(\boldsymbol{u})[i, j] = \boldsymbol{u}[i, j]^2. \quad (8)$$

Crucially, we note that SQUARE represents a special case of MULTIPLY:

$$\text{SQUARE} = \text{MULTIPLY}(0, 0, D) \quad (9)$$

so this result implies that single-layer linear attention is not expressive enough to exactly implement MULTIPLY.

# 4. Gated convolutions can precisely solve least squares

Motivated by the finding that Transformers struggle to precisely implement linear algebra operations (Section 3.2), we investigate whether an alternative architecture might improve the precision of our models. We focus on BASECONV,

a *gated convolutional model*, as a natural choice since recent work has shown they can exactly and efficiently implement arithmetic circuits (Arora et al., 2023; 2024). In Section 4.1, we recap the equivalence of gated convolutions and arithmetic circuits, and consider the more general problem of approximating smooth functions in Section 4.2. Our key observation is that gated convolutions can exactly implement polynomial activation functions (Theorem 4.2). We use this fact, plus results from approximation theory, to argue that BASECONV can efficiently approximate *smooth multivariate functions* in our main theoretical result (Theorem 4.4). In Section 4.3, we provide explicit weight constructions to argue that gated convolutions are expressive enough to solve least squares problems to high precision by directly implementing gradient descent. For the special case of linear regression, we validate our weight constructions empirically and demonstrate that gated convolutions can obtain machine precision solutions in practice.

## 4.1. Equivalence of gated convolutions and arithmetic circuits

We start by recounting prior work proving the equivalence between gated convolutions and arithmetic circuits. Throughout the paper, we focus on the BASECONV architecture (Arora et al., 2023), parameterized as in Equation 2. Since BASECONV is asymptotically equivalent to general gated convolutional models, our theoretical results directly apply to this wider class of architectures as well.

**Theorem 4.1** (Theorem H.21 from (Arora et al., 2023)). *Any depth-$\Delta$ and width-$w$ arithmetic circuit $\mathcal{C}$, and input $\mathbf{u}^{N \times D}$ can be implemented by a BASECONV model with $O(\Delta \log w)$ layers and $O(wD)$ parameters per layer.*

In particular, we note that the linear algebra primitives we specify in Section 3.2 (READ, AFFINE, and MULTIPLY) are

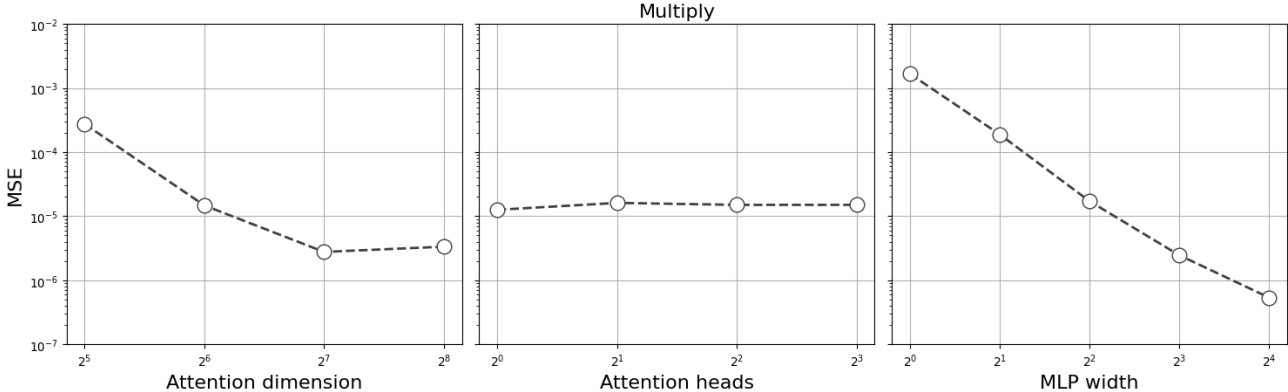

*Figure 4.* Precision of (2-layer) Transformers on MULTIPLY task scales poorly with attention dimension (left), number of heads (middle), and MLP width (right, where MLP hidden dimension = width $\times$ attention dimension).

each arithmetic circuits with depth $\Delta = O(1)$ and width $w = O(D)$. Thus, Theorem 4.1 implies there exist efficient BASECONV implementations for all three primitives. In Appendix D.2.1, we provide explicit constructions of single-layer BASECONV models that exactly implement the READ, AFFINE, and MULTIPLY primitives.

## 4.2. Approximating general smooth functions using BASECONV

In this section, we broaden our scope beyond arithmetic circuits and theoretically investigate the ability of gated convolutions to approximate the general class of smooth functions. Our key theoretical result is Theorem 4.4, which provides upper bounds on the number of layers and parameters required to $\epsilon$-approximate any multivariate smooth function $f : [-1, 1]^{N \times D} \to \mathbb{R}^{N \times D}$.

We start by noting that gated convolutions are expressive enough to represent polynomials, one of the key elements of modern approximation theory.

**Theorem 4.2.** *Given any degree-$d$ polynomial $P(X)$ and $\mathbf{u} \in [-1, 1]^{N \times D}$, there exists a BASECONV model with $O(d)$ layers and $O(ND)$ parameters per layer that exactly implements $P(\mathbf{u})$, where $P$ is applied element-wise.*

The ability to efficiently represent polynomials is crucial because polynomials form a natural and well-studied function basis. It is well known that polynomials are dense in the space of continuous functions on bounded intervals (De Branges, 1959). Modern approximation theory provides precise theoretical results about the difficulty of approximating smooth functions using polynomials:

**Theorem 4.3** (Jackson's Theorem (Pleśniak, 2009)). *Any $r$-times differentiable function $f(x) : [-1, 1] \to \mathbb{R}$ satisfying $||\frac{d^r}{dx^r} f(x)||_\infty \le L$ is $\epsilon$-approximable by a $d$-degree polynomial, where $d = O\left(\left(\frac{L}{\epsilon}\right)^{1/r} + r\right)$.*

Combining these two results, we can show that gated convolutions are able to approximate any *univariate* smooth function (Theorem D.37). Intuitively, we first approximate the function using a polynomial expansion, then use BASECONV to efficiently implement the polynomial.

Our main theoretical result, detailed in Proposition D.43, generalizes to the case of *multivariate* smooth functions:

**Theorem 4.4.** *Let $f : [-1, 1]^{N \times D} \to \mathbb{R}^{N \times D}$ be a $k$-times differentiable multivariate function. Then for all $\epsilon > 0$, there exists a BASECONV model with $O(d \log(ND))$ layers and $O((ND)^d)$ parameters that $\epsilon$-approximates $f$, where $d = O_k\left(\sqrt[k]{\frac{NDL}{\epsilon}}\right)$.*

Please see Appendix D.4 for proofs and further discussion.

## 4.3. Weight constructions: BASECONV can implement gradient descent for linear regression

For the special case of linear regression, we observe that an iteration of gradient descent can be expressed exactly as an arithmetic circuit. Thus, Theorem 4.1 implies that there exists a BASECONV model that exactly implements a gradient descent iteration on linear regression.

Concretely, we provide two $O(1)$-layer weight constructions for gradient descent using BASECONV in Appendix D.3.1. One requires a $O(D)$ state size using a *non-causal* model (i.e. each entry can access any other entry of the sequence) and one requires a $O(D^2)$ state size using a *causal* model (i.e. entries cannot access later entries of the sequence). In Appendix D.3.2, we prove that both constructions are asymptotically optimal with respect to state size.

**Empirical implementation.** To investigate the feasibility and numerical properties of our weight constructions in practice, we implement gradient descent with BASECONV as detailed in Appendix D.3.1. In Figure 5, we evaluate

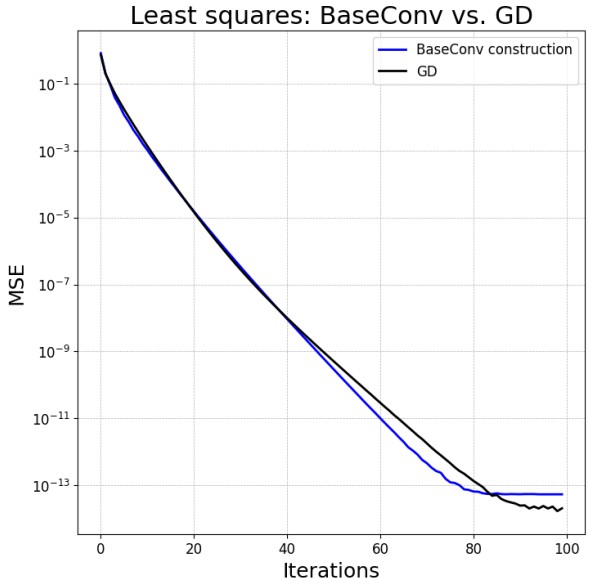

*Figure 5.* Implementation of the BASECONV weight construction for gradient descent on linear regression (Appendix D.3.1). BASECONV is expressive enough to solve least squares to high precision.

how precision scales with the effective number of layers and compare to a manual implementation of gradient descent.

We confirm empirically that BASECONV is expressive enough to algorithmically solve linear regression, matching the gradient descent iterates to high precision. Our constructed BASECONV achieves an MSE of $10^{-14}$, a lift of 9 orders of magnitude from the error saturation threshold of trained Transformers.

# 5. Towards learning high-precision algorithms

Having shown that gated convolutions theoretically close the expressivity gap on numerical algorithms in Section 4, in this section we investigate *learning* high-precision solutions.

We observe that despite our insights about the expressivity of BASECONV, they perform 2 orders of magnitude *worse* than Transformers when trained naively on in-context least squares (Figures 6, 10) and 10 orders of magnitude worse than our weight construction.

To tease apart the complexity of learning algorithmic solutions, we investigate two simplified in-context least squares training setups (Section 5.1). In both settings, we show promising results towards learning general high-precision algorithms, including precision lifts of 2-3 orders of magnitude on in-context linear regression. However, we identify the optimizer as a key bottleneck in high-precision regimes. In Section 5.2, we empirically investigate the expressivity-

learnability gap for BASECONV, using our linear algebra primitives from Section 3.2 as a natural testbed.

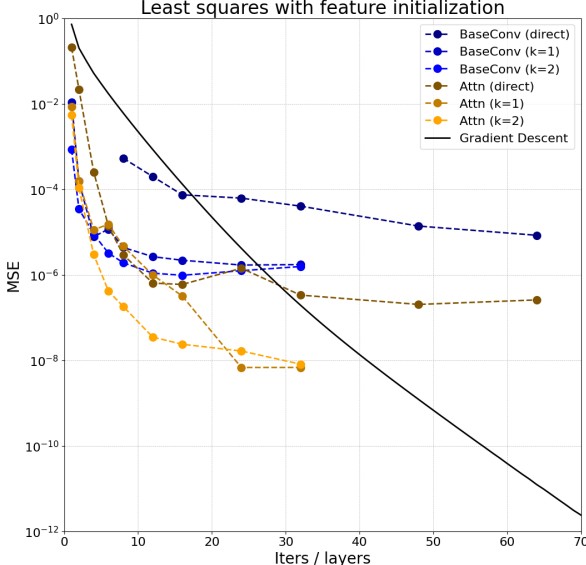

*Figure 6.* Transformers vs. BASECONV on in-context linear regression. Adding features from causal gradient descent construction (Appendix D.3.1) boosts precision by 2 orders of magnitude, though neither model has linear convergence like gradient descent.

## 5.1. Investigations on simplified in-context least squares

To better probe bottlenecks to learning high-precision algorithms, we define two simplified variants of the in-context least squares problem:

1. We append additional features to the inputs of the models, based on our causal gradient descent construction from Appendix D.3.1.

2. We explicitly supervise the models on gradient descent iterates. During inference, we apply our models iteratively, starting from a random initial guess $\boldsymbol{w}_0$, until approximate convergence to a fixed point $\boldsymbol{w}_{\inf}$.

### 5.1.1. IN-CONTEXT LEAST SQUARES WITH FEATURE INITIALIZATION

**Experiment setup.** In this experiment, we append additional features to the inputs of the models. We define three variants of the in-context least squares task, $\{LS_{init}^k\}_{k=0}^2$. In the $k$-th variant, the extra features we append to the inputs correspond to the outputs of the first $k$ layers of the causal gradient descent construction of Appendix D.3.1. Specifically, for the $k$-th variant, the $i$-th in-context example, as inputted into the model, is:

• $k = 0$ (standard in-context least squares): $\{(\boldsymbol{x}_i, y_i)\}$.

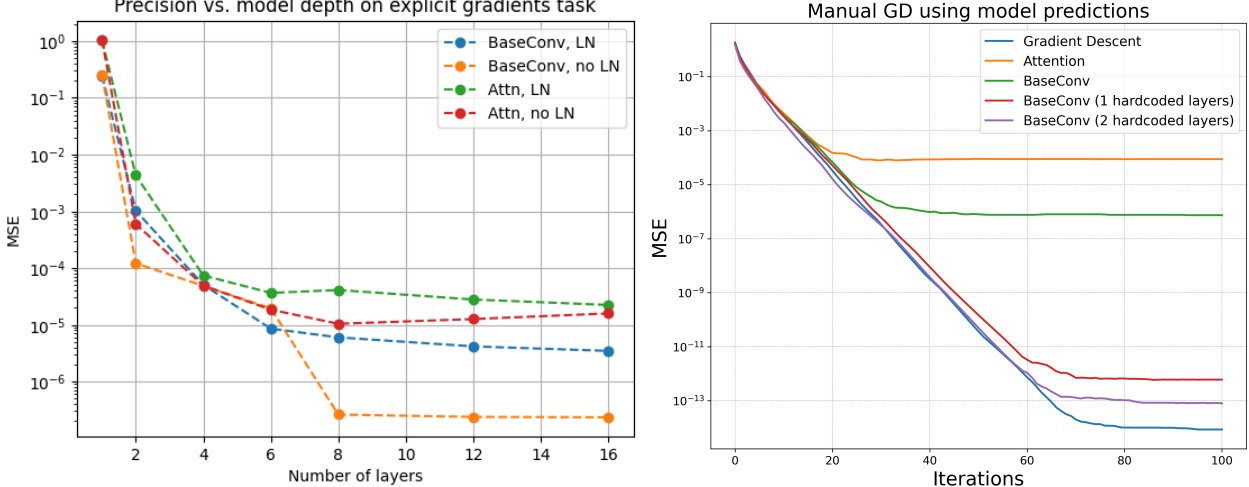

*Figure 7.* Left: Training explicitly to learn gradient descent iterates, precision of BASECONV without LayerNorms outscales Transformers. Right: Using predicted iterates to manually implement gradient descent, BASECONV saturates 2 orders of magnitude higher precision than Transformers (though neither reach machine precision). Interestingly, even BASECONV with a single hardcoded layer (red) achieves an MSE of $O(10^{-13})$.

- $k = 1$: $\left\{ \left( \boldsymbol{x}_i, y_i, y_i \boldsymbol{x}_i, \boldsymbol{x}_i \boldsymbol{x}_i^T \right) \right\}$.

- $k = 2$: $\left\{ \left( \boldsymbol{x}_i, y_i, \left( \sum_{i=1}^N y_i \boldsymbol{x}_i \right), \left( \sum_{i=1}^N \boldsymbol{x}_i \boldsymbol{x}_i^T \right) \right) \right\}$.

During training, we supervise the models in-context to predict $\boldsymbol{w}^*$. We note that for the variants $k \in \{1, 2\}$, each iteration of gradient descent can be implemented exactly using sequence-wise and element-wise sums alone (via Appendix D.3.1). Since we know that both Transformers and BASECONVs can express the READ and AFFINE primitives to high precision, we expect both classes of models should be able to implement gradient descent explicitly to solve least squares precisely. Please refer to Appendix B.2.4 for more training details.

**Evaluation.** In Figure 6, we evaluate the performance of BASECONVs and Transformers on the three tasks $\{LS_{init}^k\}_{k=0}^2$. We observe that providing additional features ($k = 1, 2$) boosts precision of both classes of models by 2 orders of magnitude, compared to the standard in-context least squares task ($k = 0$). However, we note that the convergence rate still saturates more than 7 orders of magnitude above machine precision, and neither learned model matches the linear convergence rate of gradient descent. This gap between *expressivity* and *learnability* suggests the optimizer remains a key bottleneck for learning high-precision algorithms.

### 5.1.2. EXPLICITLY LEARNING GRADIENT UPDATES

**Experiment setup.** In this experiment, we explicitly supervise Transformers and BASECONVs to predict the gradi-

ent of the least squares objective, Equation 12:

$$\{(\boldsymbol{x}_1, y_1), \ldots, (\boldsymbol{x}_N, y_N), \boldsymbol{w}_0\} \to \nabla \mathcal{L}(\boldsymbol{w}_0). \qquad (10)$$

During inference, we apply our models iteratively, using our model predictions to explicitly perform gradient descent. Starting from a random guess $\boldsymbol{w}_0$, we repeatedly compute:

$$T_\theta \left( \{(\boldsymbol{x}_1, y_1), \ldots, (\boldsymbol{x}_N, y_N), \boldsymbol{w}_i\} \right) := \boldsymbol{\Delta}_i \qquad (11)$$

and define $\boldsymbol{w}_{i+1} := \boldsymbol{w}_i - \eta \boldsymbol{\Delta}_i$. until approximate convergence to a fixed point $\boldsymbol{w}_\infty$. We compare to the true $\boldsymbol{w}^*$. Refer to Appendix B.2.5 for more details.

**Evaluation.** We evaluate the performance of our setup in Figure 7. Increasing the model depth, we find that BASEC-ONVs *without LayerNorms* outperform Transformers on learning the explicit gradient descent circuit (a gap of 2 orders of magnitude for our largest models.)

Next, we train 3-layer Transformers and BASECONVs and evaluate the performance of the models applied as *iterative* algorithms. Excitingly, we find that the learned models are robust enough that iterates continue to converge even after 40+ iterations. As with the gradient, we observe that the BASECONV model outperforms the parameter-matched Transformer by 2 orders of magnitude. However, its precision ($O(10^{-6})$) is still 8 orders of magnitude worse than our weight construction (Appendix D.3.1).

**Explicit gradient descent with feature initialization.** Finally, we compare to 3-layer BASECONVs whose first $k \in \{1, 2\}$ layers are *frozen* and initialized to the weight construction. Note that this is equivalent to applying the feature

initialization technique from Section 5.1.1. We observe a 6 orders of magnitude expressivity-learnability gap between the partially frozen and fully trained BASECONVs. Notably, the BASECONV models with feature initialization reach near machine precision, closely matching the performance of true gradient descent on this problem. These findings show that even learning the 3-layer arithmetic circuit representing an iterate of gradient descent to high precision remains challenging.

## 5.2. Investigating the expressivity-learning gap with MULTIPLY

To further probe the precision bottlenecks introduced by the optimizer, we train BASECONV models on the simple MULTIPLY task from Section 3.2. In Figure 8, we scale BASECONV's size, demonstrating that although BASEC-ONVs train to $O(10^{-9})$ on this task, they struggle to achieve machine precision solutions. In Figure 9, we increase training time, demonstrating that BASECONV precision on the MULTIPLY task improves steadily but precision gains diminish exponentially. We highlight the difficulty of reaching machine precision solutions even on the simplest expressible tasks, and leave this challenge to future work.

## 6. Conclusion

In this work, we explore the capabilities of Transformers to solve high-precision numerical problems. Surprisingly, we demonstrate that Transformers struggle to solve least squares to high precision even on noiseless fully-determined problems. We investigate gated convolutions as one way of getting to high-precision algorithms, showing that these models can precisely solve least squares by explicitly implementing gradient descent. We propose a simple training setup for explicitly learning gradient descent, with which we demonstrate an improvement of 2 orders of magnitude upon in-context Transformers. However, we highlight the optimizer as a key bottleneck in high-precision regimes, which we leave for future work.

## Acknowledgements

We thank Yasa Baig, Aman Timalsina, Sabri Eyuboglu, Michael Zhang, Dylan Zinsley, and Benjamin Spector for their helpful feedback and discussion during this work.

We gratefully acknowledge the support of NIH under No. U54EB020405 (Mobilize), NSF under Nos. CCF2247015 (Hardware-Aware), CCF1763315 (Beyond Sparsity), CCF1563078 (Volume to Velocity), 1937301 (RTML), and PHY-2019786 (The NSF AI Institute for Artificial Intelligence and Fundamental Interactions, http://iaifi.org/); US DEVCOM ARL under Nos. W911NF-23-2-0184 (Long-context) and W911NF-21-2-0251 (Interactive Human-AI Teaming); ONR under Nos. N000142312633 (Deep Signal Processing); Stanford HAI under No. 247183; U.S. Department of Energy, Office of Science, Office of Advanced Scientific Computing Research, Department of Energy Computational Science Graduate Fellowship under Award Number DE-SC0023112; NXP, Xilinx, LETI-CEA, Intel, IBM, Microsoft, NEC, Toshiba, TSMC, ARM, Hitachi, BASF, Accenture, Ericsson, Qualcomm, Analog Devices, Google Cloud, Salesforce, Total, the HAI-GCP Cloud Credits for Research program, the Stanford Data Science Initiative (SDSI), and members of the Stanford DAWN project: Meta, Google, and VMWare. The U.S. Government is authorized to reproduce and distribute reprints for Governmental purposes notwithstanding any copyright notation thereon. Any opinions, findings, and conclusions or recommendations expressed in this material are those of the authors and do not necessarily reflect the views, policies, or endorsements, either expressed or implied, of NIH, ONR, or the U.S. Government. JG and AR's research is supported by NSF grant CCF#2247014.

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

# Appendix

The appendix is organized as follows:

- Appendix A provides a more detailed overview of related work.

- Appendix B provides details about our experimental setup.

- Appendix C provides additional ablation studies.

- Appendix D provides details about our main theoretical results.

# A. Extended background

## A.1. Linear regression

Linear regression, where $f(\boldsymbol{x}; \boldsymbol{w}) = \boldsymbol{w}^T \boldsymbol{x}$, is an important class of least squares problems. Linear regression is well-understood theoretically, and we know of simple numerical algorithms for solving linear regression to high precision (Weisberg, 2005; Boyd & Vandenberghe, 2004). We focus on two algorithms: gradient descent and Newton's iteration.

**Gradient descent** Given a guess for $\boldsymbol{w}^*$, we minimize the least squares loss

$$\mathcal{L}(\boldsymbol{w}) = \frac{1}{2N} \sum_{i=1}^{N} (f(\boldsymbol{x}_i; \boldsymbol{w}) - y_i)^2 \tag{12}$$

via gradient descent on $\boldsymbol{w}$:

$$\nabla_{\boldsymbol{w}} \mathcal{L}_N = \frac{1}{N} \sum_{i=1}^{N} (\boldsymbol{w}^T \boldsymbol{x}_i - y_i) \boldsymbol{x}_i \tag{13}$$

$$\boldsymbol{w}_{t+1} = \boldsymbol{w}_t - \eta \nabla \mathcal{L}_N(\boldsymbol{w}_t) \tag{14}$$

**Ordinary Least Squares and Newton's iteration** In the noiseless, full determined regime, the Bayes-optimal estimator is ordinary least squares (OLS) (Weisberg, 2005):

$$\boldsymbol{w}^{OLS} = (\boldsymbol{X}^T \boldsymbol{X})^{-1} \boldsymbol{X}^T \boldsymbol{y}, \tag{15}$$

where

$$\boldsymbol{X} = \begin{pmatrix} \leftarrow \boldsymbol{x}_1 \rightarrow \\ \vdots \\ \leftarrow \boldsymbol{x}_N \rightarrow \end{pmatrix}, \quad \boldsymbol{y} = \begin{pmatrix} y_1 \\ \vdots \\ y_N \end{pmatrix} \tag{16}$$

Note that this estimator requires a matrix inverse, which is expensive to compute exactly. An alternative is to use Newton's method to approximate the matrix inverse term (Schulz, 1933). To estimate $(\boldsymbol{X}^T \boldsymbol{X})^{-1}$, we can perform the following iterative algorithm:

$$\boldsymbol{A}_{t+1} = \boldsymbol{A}_t (2\boldsymbol{I} - (\boldsymbol{X}^T \boldsymbol{X}) \boldsymbol{A}_t) \tag{17}$$

where $\boldsymbol{A}_t$ converges to $(\boldsymbol{X}^T \boldsymbol{X})^{-1}$.

## A.2. Related work

In this section, we detail prior work on in-context learning, Transformer expressivity, and gated convolutional architectures.

**In-context learning.** The capability of Transformers to perform in-context learning on language and pattern matching tasks has been well-documented (Brown et al., 2020; Dasgupta et al., 2022; Wei et al., 2022). More recently, a flurry of work has investigated in-context learning for regression-style tasks. (Garg et al., 2022) first formulated the mathematical framework to analyze the estimators Transformers implement in-context, focusing on linear regression and other least squares problems. A number of works further observed empirically that Transformers seem to approximate Bayes-optimal estimators on distributional problems. For example, based on the task distribution, the performance of in-context Transformers mimics optimally-tuned LASSO on sparse linear regression, ridge regression on noisy dense linear regression, and Bayes-optimal priors for task mixtures (Akyürek et al., 2024; Raventós et al., 2024; Yadlowsky et al., 2023; Ahuja et al., 2023; Bai et al., 2024). Beyond standard least squares problems, other works have investigated the ability of Transformers to in-context solve broader problems of scientific interest like differential equations (Yang et al., 2023; Chen et al., 2024; Liu et al., 2023).

Towards explaining these observations, recent works have focused on understanding the expressivity and optimziation landscapes of Transformer variants (typically non-causal linear attention) on linear regression. Linear attention has been shown to be expressive enough to implement numerical algorithms for solving linear regression, including gradient descent (Akyürek et al., 2022; Von Oswald et al., 2023) and Newton's method (Fu et al., 2023; Giannou et al., 2024). Recent work has begun to investigate the optimization dynamics for linear attention on least squares. (Ahn et al., 2024; Mahankali et al., 2023) prove that the global minimizer of the in-context learning loss for linear regression using linear attention is

equivalent to a step of preconditioned gradient descent. Additionally, (Zhang et al., 2023) provides suitable conditions under which gradient flow provably converges to this global minimizer.

Transformers with non-linear attention are less well-understood theoretically. (Bai et al., 2024) provides constructions implementing optimization algorithms for a variety of least squares problems, including sparse linear and logistic regression, using RELU-activation attention. On the optimization front, we are aware of (Huang et al., 2023), which provides convergence guarantees for single-layer softmax attention under a structured data model.

Unlike prior work, we investigate the in-context learning capabilities of standard (multi-layer softmax-attention) Transformers, focusing on exploring their capability to perform *high-precision* optimization algorithms. Noting a gap between empirical performance and theoretical claims regarding in-context least squares as gradient descent, we further investigate alternative architectures to softmax attention.

**Expressivity and approximation ability of Transformers.** Although Transformers were initially designed for discrete tasks like language modeling, recent works have investigated the ability of the Transformer architecture to express general *continuous-valued* sequence-to-sequence maps. We briefly mention three classes of prior work:

- **Constructive arguments.** We highlight (Giannou et al., 2023), which proposes a looped-Transformer weight construction that implements a basic mathematical instruction set. Using compositions of these instructions, the authors demonstrate that Transformers are expressive enough to implement numerical algorithms, including matrix inversion and SGD on linear models.

- **Universal approximation results.** Several works, such as (Yun et al., 2020a;b), provide bounds on the number of parameters and layers required to approximate smooth sequence-to-sequence functions to arbitrary precision using Transformers.

- **Complexity theory results.** Recent works (Chiang et al., 2023; Merrill & Sabharwals, 2023; Merrill & Sabharwal, 2024) prove that *log-precision* Transformers lie in $TC^0$, a limited complexity class of circuits.

**Gated convolutions.** Gated convolutional models are a class of architectures that serve as an efficient alternative to attention. These models, consisting of gating (element-wise multiplication) and long convolutions (filter size equal to sequence length), stem from earlier work (Gu et al., 2021) inspired by the signal processing literature. In this work we focus on the BASECONV model from (Arora et al., 2023), but a recent surge of interest in efficient attention replacements has led to a flood of gated convolutional architectures (Poli et al., 2023; Peng et al., 2023; Gu & Dao, 2023).

Recent architectural innovations within the class of gated convolutional models have been largely motivated by language modeling tasks (Fu et al., 2022; Arora et al., 2023). Unlike these prior works, which focus on matching attention's performance on *discrete* tasks, we observe that the connection between gated convolutions and arithmetic circuits implies they are able to exactly express a range of important numerical algorithms for *continuous-valued* tasks. We further investigate their ability to learn these algorithms in-context.

# B. Experimental setup

Here, we provide additional details about our experimental setup.

## B.1. Model architecture

We base our Transformer and BASECONV models off the GPT2 family (Radford et al., 2019). Unless otherwise specified, we use the following default settings:

| Config | Setting |
|---|---|
| Embedding size | 64 |
| Number of layers | 12 |
| Number of heads | 1 |
| MLPs | True |
| MLP hidden size | 4x embedding size |
| MLP activation | ReLU |
| Batch size | 256 |
| Optimizer | Adam |
| Learning rate | $10^{-3}$ |
| Scheduler | StepLR |
| Training iterations | $10^6$ |
| Step rate | $10^4$ |
| Decay rate | 0.9 |
| Problem dim | 5 |
| Sequence length | 20 |

## B.2. Tasks

Each of our in-context learning tasks can be viewed as a sequence-to-sequence map

$$\mathcal{M} : \mathbb{R}^{N_{in} \times D_{in}} \to \mathbb{R}^{N_{out} \times D_{out}}$$

In this subsection, we provide details about task implementations, specifying the input/output formats for each of the synthetic tasks and in-context least squares variants we implement.

### B.2.1. IN-CONTEXT LINEAR REGRESSION, N PARALLEL TASKS.

In Figure 1, we use the in-context linear regression setup from (Garg et al., 2022), $\mathcal{M}_{LS\_parallel} : \mathbb{R}^{(2N+1) \times D} \to \mathbb{R}^{(N+1) \times 1}$, where the inputs are formatted as

$$\boldsymbol{u}_{in} := \begin{bmatrix} \boldsymbol{x}_1 & y_1 \boldsymbol{e}_1 & \dots & \boldsymbol{x}_N & y_N \boldsymbol{e}_1 & \boldsymbol{x}_{query} \end{bmatrix}$$

and the expected outputs are

$$T_\theta(\boldsymbol{u}_{in})[0::2, :1] := \begin{bmatrix} y_1 & \dots & y_N & y_{query} \end{bmatrix}.$$

### B.2.2. IN-CONTEXT LINEAR REGRESSION, FULLY-DETERMINED, FIXED N.

In Figure 2, we simplify the linear regression setup from (Garg et al., 2022) by supervising only on the final prediction $y_{query}$. Concretely, we consider $\mathcal{M}_{LS\_fixed\_N} : \mathbb{R}^{(N+1) \times (D+1)} \to \mathbb{R}$, where as above the inputs are formatted as

$$\boldsymbol{u}_{in} := \begin{bmatrix} \boldsymbol{x}_1 & \dots & \boldsymbol{x}_N & \boldsymbol{x}_{query} \\ y_1 & \dots & y_N & 0 \end{bmatrix}$$

and the expected output is

$$T_\theta(\boldsymbol{u}_{in})[\text{-1:}, \text{-1:}] := y_{query}.$$

We note that causal softmax Transformers achieve higher precision on this "fixed length" variant (compare Figure 2 to Figure 1).

**B.2.3. PRIMITIVES.**

For each of the primitives (Figures 3, 4, 8, 9), we increase the task size, setting $D = 20$ and $N = 40$.

- READ is defined as $\mathcal{M}_{Read} : \mathbb{R}^{N \times D} \to \mathbb{R}^{N \times D}$, where the inputs are formatted as

$$\boldsymbol{u}_{in} \in \mathbb{R}^{N \times D} := \begin{bmatrix} \boldsymbol{x}_1 & \dots & \boldsymbol{x}_N \end{bmatrix}$$

  and the expected outputs are $T_\theta(\boldsymbol{u}_{in}) \in \mathbb{R}^{N \times D}$ such that

$$T_\theta(\boldsymbol{u}_{in})[k, :] := \begin{cases} \boldsymbol{u}_{in}[i, :] & k = j \\ \boldsymbol{u}_{in}[k, :] & k \neq j \end{cases}$$

  for task parameters $i \neq j \in [N]$.

- AFFINE is defined as $\mathcal{M}_{Affine} : \mathbb{R}^{N \times D} \to \mathbb{R}^{N \times 1}$, where the inputs are formatted as

$$\boldsymbol{u}_{in} \in \mathbb{R}^{N \times D} := \begin{bmatrix} \boldsymbol{x}_1 & \dots & \boldsymbol{x}_N \end{bmatrix}$$

  and the expected outputs are

$$T_\theta(\boldsymbol{u}_{in}) := \begin{bmatrix} \boldsymbol{x}_1^T \boldsymbol{h} & \dots & \boldsymbol{x}_N^T \boldsymbol{h} \end{bmatrix}$$

  where $\boldsymbol{h} \in \mathbb{R}^D$ is a task parameter.

- MULTIPLY is defined as $\mathcal{M}_{Multiply} : \mathbb{R}^{N \times D} \to \mathbb{R}^{N \times D/2}$, where the inputs are formatted as

$$\boldsymbol{u}_{in} \in \mathbb{R}^{N \times D} := \begin{bmatrix} \boldsymbol{x}_1 & \dots & \boldsymbol{x}_N \end{bmatrix}$$

  and the expected outputs are

$$T_\theta(\boldsymbol{u}_{in}) := \big( \boldsymbol{x}_1[:, : D/2] \odot \boldsymbol{x}_1[:, D/2 :] \quad \dots \quad \boldsymbol{x}_N[:, : D/2] \odot \boldsymbol{x}_N[:, D/2 :] \big).$$

**B.2.4. FEATURE INITIALIZATION LINEAR REGRESSION.**

In Figure 6, we use a simplified linear regression setup, in which additional features are provided to the model, toward encouraging the model to explicitly implement gradient descent in-context. We proceed to define the task $\mathcal{M}_{LS\_feature} : \mathbb{R}^{N \times (D^2 + 2D + 1)} \to \mathbb{R}^D$.

There are three variants of the task, $k \in \{0, 1, 2\}$, which indicates that the appended features are the outputs of the $k$-th layer of the causal gradient descent construction from Appendix D.3.1. See Section 5.1.1 for more details.

For $k = 0$, the inputs are

$$\boldsymbol{u}_{in} := \begin{bmatrix} \boldsymbol{x}_1 & \dots & \boldsymbol{x}_N \\ y_1 & \dots & y_N \\ \boldsymbol{0} & \dots & \boldsymbol{0} \\ \boldsymbol{0} & \dots & \boldsymbol{0} \end{bmatrix}.$$

For $k = 1$, the inputs are

$$\boldsymbol{u}_{in} := \begin{bmatrix} \boldsymbol{x}_1 & \dots & \boldsymbol{x}_N \\ y_1 & \dots & y_N \\ y_1 \boldsymbol{x}_1 & \dots & y_N \boldsymbol{x}_N \\ flt(\boldsymbol{x}_1 \boldsymbol{x}_1^T) & \dots & flt(\boldsymbol{x}_N \boldsymbol{x}_N^T) \end{bmatrix}.$$

For $k = 2$, the inputs are

$$\boldsymbol{u}_{in} := \begin{bmatrix} \boldsymbol{x}_1 & \dots & \boldsymbol{x}_N \\ y_1 & \dots & y_N \\ \leftarrow \sum_{i=1}^N y_i \boldsymbol{x}_i \rightarrow \\ \leftarrow \sum_{i=1}^N flt(\boldsymbol{x}_i \boldsymbol{x}_i^T) \rightarrow \end{bmatrix}$$

where $flt$ denotes the `flatten` operation.

In all cases, the expected outputs are

$$T_\theta(\boldsymbol{u}_{in})[\text{-1:}, \text{:D}] := \boldsymbol{w}^*.$$

For this task, we use an embedding size of 256.

B.2.5. EXPLICIT GRADIENT UPDATES.

In Figure 7, we investigate a simple training setting, in which the model is explicitly trained to predict the gradient of the least squares loss. We proceed to define the task $\mathcal{M}_{gradient} : \mathbb{R}^{(N+1)\times(D^2+2D+1)} \to \mathbb{R}^D$.

As in the feature initialization linear regression task, we consider three variants of the task, $k \in \{0, 1, 2\}$. The inputs are similar to the previous task:

For $k = 0$, the inputs are

$$\boldsymbol{u}_{in} := \begin{bmatrix} \boldsymbol{x}_1 & \ldots & \boldsymbol{x}_N & \boldsymbol{w}_0 \\ y_1 & \ldots & y_N & 0 \\ \boldsymbol{0} & \ldots & \boldsymbol{0} & \boldsymbol{0} \\ \boldsymbol{0} & \ldots & \boldsymbol{0} & \boldsymbol{0} \end{bmatrix}.$$

For $k = 1$, the inputs are

$$\boldsymbol{u}_{in} := \begin{bmatrix} \boldsymbol{x}_1 & \ldots & \boldsymbol{x}_N & \boldsymbol{w}_0 \\ y_1 & \ldots & y_N & 0 \\ y_1\boldsymbol{x}_1 & \ldots & y_N\boldsymbol{x}_N & \boldsymbol{0} \\ flt(\boldsymbol{x}_1\boldsymbol{x}_1^T) & \ldots & flt(\boldsymbol{x}_N\boldsymbol{x}_N^T) & \boldsymbol{0} \end{bmatrix}.$$

For $k = 2$, the inputs are

$$\boldsymbol{u}_{in} := \begin{bmatrix} \boldsymbol{x}_1 & \ldots & \boldsymbol{x}_N & \boldsymbol{w}_0 \\ y_1 & \ldots & y_N & 0 \\ \leftarrow \sum_{i=1}^{N} y_i\boldsymbol{x}_i \rightarrow \\ \leftarrow \sum_{i=1}^{N} flt(\boldsymbol{x}_i\boldsymbol{x}_i^T) \rightarrow \end{bmatrix}$$

where $flt$ denotes the `flatten` operation.

In all cases, the expected outputs are

$$T_\theta(\boldsymbol{u}_{in})[\text{-1:}, \text{:D}] := \nabla_{\boldsymbol{w}}\mathcal{L}(\boldsymbol{w}_0).$$

For this task, we use an embedding size of 256.

## B.3. Data generation

At each training step, we produce a random training prompt $\boldsymbol{u}_{in}$ by sampling each variable randomly: from the isotropic Gaussian distribution $N(\boldsymbol{0}, \boldsymbol{I})$ for continuous-valued parameters, and from the uniform distribution for discrete parameters. Concretely:

- For the in-context linear regression tasks, input vectors $\boldsymbol{x}_1, \ldots, \boldsymbol{x}_N$ are sampled from $N(\boldsymbol{0}^D, \boldsymbol{I}^D)$, and the unknown linear function is determined by $\boldsymbol{w}^*$, also drawn from $N(\boldsymbol{0}^D, \boldsymbol{I}^D)$.

- For the synthetic tasks READ, AFFINE, MULTIPLY (Section 3.2), *each column* of the inputs $\boldsymbol{u}_{in} \in \mathbb{R}^{N \times D}$ is sampled from the isotropic Gaussian distribution $N(\boldsymbol{0}^D, \boldsymbol{I}^D)$. The tasks READ and AFFINE require specifying additional parameters as follows:
  - For READ, at each iteration, $i \neq j \in [N]$ are sampled uniformly.
  - For AFFINE, at each iteration, the affine transformation $\boldsymbol{h}$ is sampled from $N(\boldsymbol{0}^D, 3\boldsymbol{I}^D)$.

- For the explicit gradient task, the random initialization $\boldsymbol{w}_0$ is also drawn from $N(\boldsymbol{0}^D, \boldsymbol{I}^D)$.

The model is trained to minimize the in-context training loss (Equation 4), equivalent to minimizing mean squared error over the distribution of prompts.

## C. Additional experimental results

### C.1. Primitives: Transformer vs. BASECONV

In Figure 8, we train Transformers and BASECONVs, with and without LayerNorms (LN), on the READ, AFFINE, and MULTIPLY primitives from Section 3.2. We vary the model depth $L \in \{1, 2, 4, 8\}$ and investigate how precision scales with number of layers.

We show that Transformers and BASECONVs both achieve high precision ($< O(10^{-9})$) on the READ and AFFINE tasks. However, the Transformers struggle to implement MULTIPLY to high precision, and performance scales poorly with model depth.

We observe that BASECONV without LayerNorm generally performs the best across all three primitives, consistently outperforming BASECONV with LayerNorm by 2-4 orders of magnitude. Interestingly, we also find that none of the models reach machine precision ($O(10^{-15})$ for single-precision training) on these tasks. This suggests that optimizing to machine precision, even on simple tasks with no expressivity gap, remains a challenge.

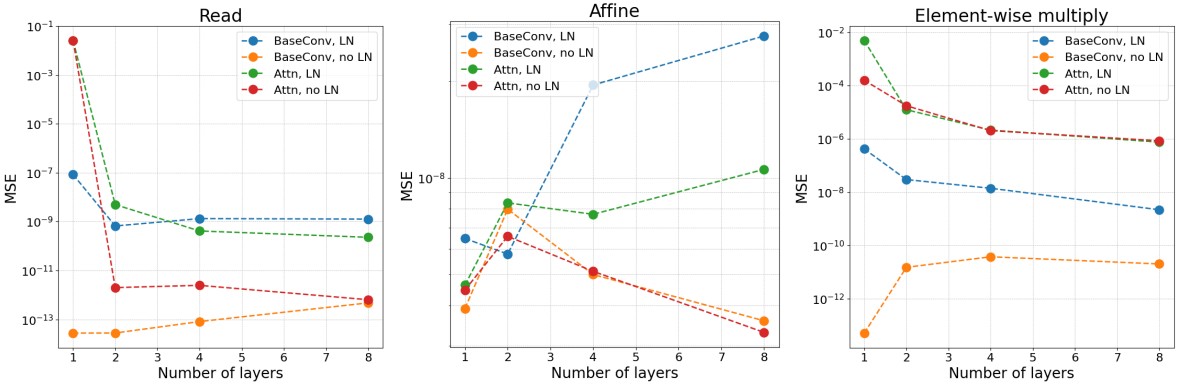

*Figure 8.* Attention vs. BASECONV on synthetic tasks. Precision consistently scales better with depth for BASECONV models than for Transformers. READ and AFFINE tasks to high precision, precision scales poorly for the MULTIPLY task.

### C.2. Scaling model training duration

In Figure 9, we train 1-layer Transformers and BASECONVs (with LayerNorms) on the MULTIPLY primitive (Section 3.2). We vary the number of iterations for which the model is trained. Recall that since new data is sampled at each iteration, we also effectively scale the dataset size proportionally. To keep the learning rates consistent across runs, we scale back the scheduler step size accordingly:

$$num\_iters \in \{10^5, 10^6, 10^7, 10^8\}$$
$$step\_size \in \{10^3, 10^4, 10^5, 10^6\}$$

We observe a power law, particularly clearly for BASECONV, as we scale from $10^5$ to $10^8$ iterations. Both models achieve a 2-3 order of magnitude improvement in precision as we increase training duration by 3 orders of magnitude. We leave it to future work to investigate whether it is possible to scale precision more efficiently using more refined optimization methods.

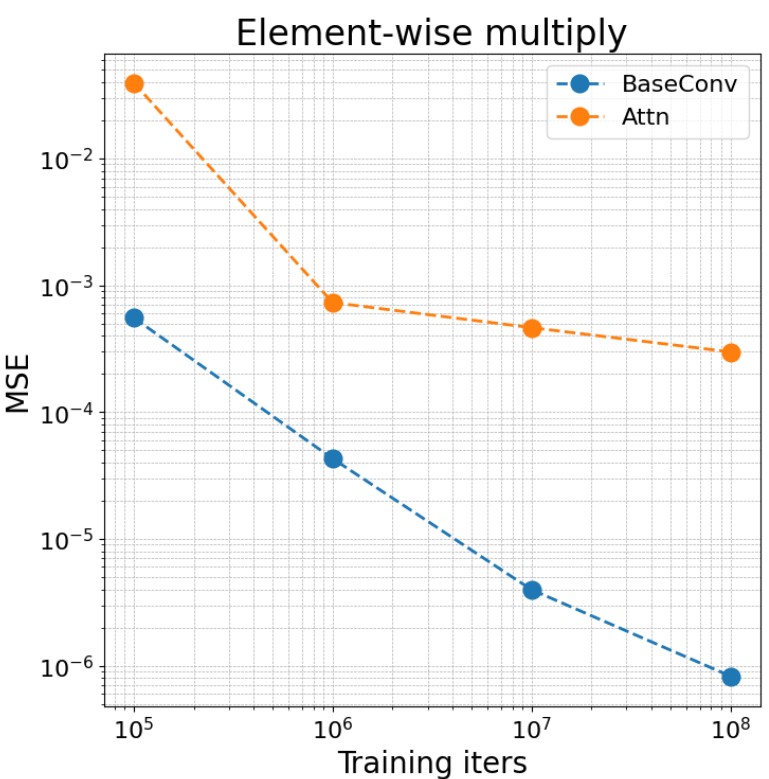

*Figure 9.* Scaling number of training iterations for 1-layer Transformer vs. BASECONV on the MULTIPLY task. Both models improve precision by 2-3 orders of magnitude as training duration increases by 3 orders of magnitude.

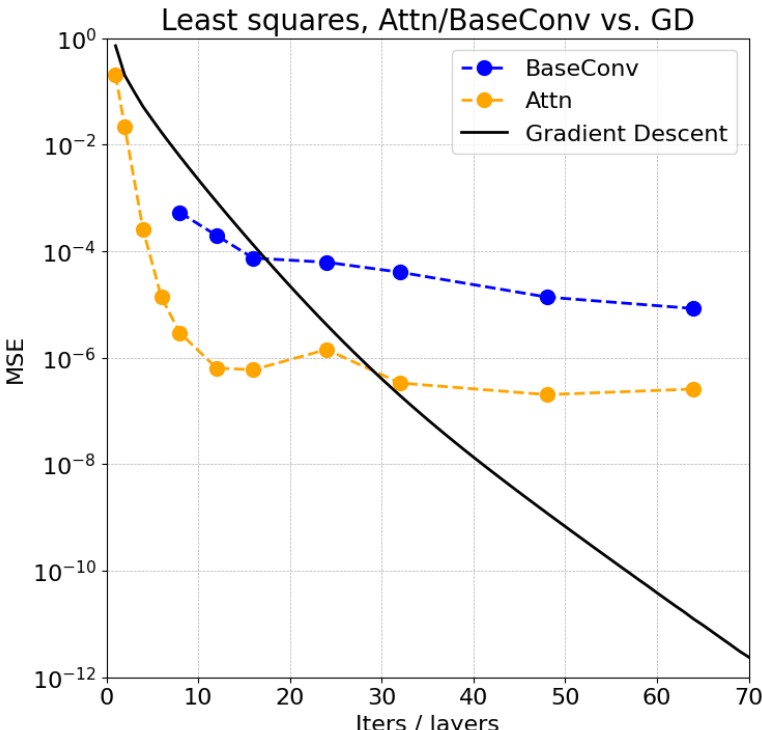

*Figure 10.* Transformers vs. BASECONVs trained on (*fixed N*) fully-determined least squares. Despite empirical constructions demonstrating that BASECONVs can solve least squares to high precision by implementing gradient descent, *learned* BASECONV models scale *worse* than learned Transformers: a difference of 2 orders of magnitude for the largest models.

# D. Theoretical results

This section is organized as follows:

- We detail notation and definitions in Appendix D.1.

- In Appendix D.2, we include theoretical results regarding the primitives from Section 3.2: expressivity results with BASECONV and attention, and iterative algorithms as compositions of primitives.

- In Appendix D.3, we discuss upper and lower bounds for implementing gradient descent on least squares using BASECONV, supplementing Section 4.1.

- In Appendix D.4, we provide missing theoretical details from Section 4.2 regarding BASECONV and polynomials.

## D.1. Notation

We heavily borrow notation from Appendix H of (Arora et al., 2023), which we recollect below. We denote the all 1 row vector of size $k$, given by $\begin{bmatrix} 1 & 1 & \ldots & 1 & 1 \end{bmatrix}$, and the all 0 row vector of size $k$, given by $\begin{bmatrix} 0 & 0 & \ldots & 0 & 0 \end{bmatrix}$, as $\mathbf{1}^k$ and $\mathbf{0}^k$, respectively. We also construe the standard basis vector $\mathbf{e}_i$ as a column vector in this appendix, and adhere to the following matrix indexing convention: $\mathbf{M}[i, j]$ is the entry in the $i$th row and the $j$th column, $\mathbf{M}[i, :] \in \mathbb{F}^{1 \times n}$ denotes the $i$th row, and $\mathbf{M}[:, j] \in \mathbb{F}^{m \times 1}$ denotes the $j$th column of $\mathbf{M} \in \mathbb{F}^{m \times n}$, where $\mathbb{F}$ is a field (the reader can assume that $\mathbb{F}$ is the field of real numbers i.e. $\mathbb{F} = \mathbb{R}$). We then use $\mathbf{1}^{m \times n}, \mathbf{0}^{m \times n} \in \mathbb{F}^{m \times n}$ to denote the matrix of all 1s and 0s, respectively. We note that some notation differs from those used in earlier sections.

Next, we denote the *Hadamard product* of vectors $\mathbf{u}, \mathbf{v} \in \mathbb{F}^n$ as $\mathbf{u} \odot \mathbf{v}$; the operation can be extended to matrices by applying the Hadamard product column-wise across the matrices. This is commonly referred to as *(element-wise) gating*. For vectors $\mathbf{u}, \mathbf{v} \in \mathbb{F}^n$, we also denote their *linear (or acyclic) convolution* as $\mathbf{u} * \mathbf{v}$ and *cyclic convolution* as $\mathbf{u} \circledast \mathbf{v}$.

**Polynomial Notation.** Since convolution is equivalent to operations on polynomials, it is convenient to use them to discuss the inputs and outputs of gated convolution models. Let us define maps $\mathrm{poly} : \mathbb{F}^n \to \mathbb{F}[X]/(X^n)$ such that

$$\mathrm{poly}(\boldsymbol{u}) = \sum_{i=0}^{n-1} \boldsymbol{u}[i] X^i.$$

This allows us to map between vectors and polynomial. Accordingly, we also define $\mathrm{coeff} : \mathbb{F}[X]/(X^{n+1}) \to \mathbb{F}^n$ as the map converting polynomials back to vectors: $\mathrm{coeff}(\boldsymbol{u}(X)) = \boldsymbol{u}$ with $\boldsymbol{u}[i]$ defined as the coefficient in $\boldsymbol{u}(X)$ at degree $i$.

These operations allow us to interpret the convolution of vectors in terms of polynomial multiplication (Heideman & Burrus, 1988). More specifically, we have

$$\boldsymbol{u} * \boldsymbol{v} = \mathrm{coeff}\left(\boldsymbol{u}(X) \cdot \boldsymbol{v}(X) \mod X^n\right)$$

The following notation for a polynomial will be used in this section:

**Definition D.1.** A polynomial $P(X)$ with degree $d$ and some coefficients $\mathbf{c} \in \mathbb{R}^{d+1}$ is defined as,

$$P(X) = \sum_{i=0}^{d} c_i X^i.$$

Further, the degree of $P(X)$ will be denoted as $\deg(P)$.

**Function Approximation.** In this part, we collect notation and known results about function approximation. We will reference some definitions from (Pleśniak, 2009; Petersdorff, 2015; Smoothness, 2006).

The following notation is to denote the $k$th derivative of a function:

**Definition D.2.** For some function $f : \mathbb{R} \to \mathbb{R}$, $f^{(k)} := \frac{d^k}{dx^k} f(x)$ is the $k$th derivative of $f$.

Define a set of univariate functions with a notion of continuity:

**Definition D.3.** We denote $C^k[a, b]$ for $k = 1, 2, \ldots$ the space of univariate functions $f : [a, b] \to \mathbb{R}$, which have derivatives $f^{(1)}, \ldots, f^{(k)}$ that are continuous on the closed interval $[a, b]$.

Next we define a set of multivariate functions with a notion of continuity:

**Definition D.4.** A function $f : [a, b]^n \to \mathbb{R}$ is in $C^k[a, b]^n$ for $k = 1, 2, \ldots$ if all partial derivatives

$$\frac{\partial^\alpha}{\partial x_1^{\alpha_1} \partial x_2^{\alpha_2} \cdots \partial x_n^{\alpha_n}} f(y_1, y_2, \ldots, y_n)$$

exist and are continuous, for every $\alpha_1, \alpha_2, \ldots, \alpha_n \in \mathbb{Z}_{\geq 0}$, such that $\alpha_1 + \alpha_2 + \cdots + \alpha_n \leq k$ and every $(y_1, \ldots y_n) \in [a, b]^n$.

We use the following notation for the set of all univariate polynomials:

**Definition D.5.** For any integer $d \geq 0$, we define

$$\mathcal{P}_d(X) = \{c_0 + c_1 X + \cdots + c_d X^d | c_k \in \mathbb{R}\}.$$

In other words, $P_d(X)$ is the space of univariate polynomials of degree less or equal to $d$.

We use the following notation for multivariate polynomials:

**Definition D.6.** For any integers $n, d \geq 0$ , we define

$$\mathcal{P}_d^n(X_1, \ldots, X_n) = \left\{ \sum_{\boldsymbol{\alpha} = (\alpha_1, \ldots, \alpha_n) \in \mathbb{Z}_{\geq 0}^n} c_\alpha X_1^{\alpha_1} X_2^{\alpha_2} \cdots X_n^{\alpha_n} \Bigg| c_\alpha \in \mathbb{R}, \sum_{i=0}^n \alpha_i \leq d \right\}.$$

Then $\mathcal{P}_d^n(X_1, \ldots X_n)$ is the space of $n$-variate polynomials of degree less or equal to $d$.

The following notation is for considering the pointwise absolute value of a matrix:

**Definition D.7.** For $M \in \mathbb{R}^{N \times D}$ define,

$$\|M\|_\infty = \max_{\substack{0 \leq i < N \\ 0 \leq j < D}} |M[i, j]| .$$

Now lets define the corresponding $\infty-$norm for functions:

**Definition D.8.** For $g : [-1, 1]^{N \times D} \to \mathbb{R}^{N \times D}$, define

$$\|g\|_\infty = \max_{\mathbf{x} \in [-1,1]^{N \times D}} |g(\mathbf{x})| .$$

We will use the following version of Jackson's theorem for univariate inputs:

**Theorem D.9** ((D. Jackson, 1930) Jackson's Theorem for $C^k[-1, 1]$.)**.** *Let $d, k$ be integers with $d + 1 \geq k \geq 0$ and $f \in C^k[-1, 1]$. Then*

$$\inf_{P \in \mathcal{P}_d} \|f - P\|_\infty \leq \left(\frac{\pi}{2}\right)^k \frac{1}{(d+1)d \cdots (d-k+2)} \left\|f^{(k)}\right\|_\infty . \tag{18}$$

We will use the following version of Jackson's theorem for multivariate inputs:

**Theorem D.10** ((Pleśniak, 2009) Jackson's Theorem for $C^k[-1, 1]^n$.)**.** *Let $d, k$ be integers with $d + 1 \geq k \geq 0$ and $f \in C^k[-1, 1]^n$. Then*

$$\inf_{P \in \mathcal{P}_d^n} \|f - P\|_\infty \leq \frac{c_k}{d^k} \sum_{j=1}^n \left\| \frac{\partial^{k+1}}{\partial x_j^{k+1}} f(\mathbf{x}) \right\|_\infty \tag{19}$$

*where $c_k$ is a positive constant.*

We will use the following definition of univariate smooth functions:

**Definition D.11.** We call a $k$ times differentiable function $f : [-1, 1] \to \mathbb{R}$ to be $(k, L)$-smooth if $\left\|f^{(k)}\right\|_\infty \leq L$.

Next, we observe that given a univariate smooth function, there's a univariate bounded degree polynomial that approximates it to some error, $\epsilon$:

**Corollary D.12.** *For some $(k, L)$-smooth univariate function $f$ (as in Definition D.11), then there exists a polynomial $P_f(x)$ with*

$$\deg(P_f) \leq O\left( \sqrt[k]{\frac{L}{\epsilon}} \right) + k$$

*such that for all $x \in [-1, 1]$*

$$|f(x) - P_f(x)| \leq \epsilon.$$

*Proof.* We will be a bit more specific on an upper bound of $\deg(P_f)$. We pick:

$$\deg(P_f) = \left\lceil \frac{\pi}{2} \left( \frac{L}{\epsilon} \right)^{\frac{1}{k}} + k \right\rceil. \tag{20}$$

Let $d = \deg(P_f)$ where $P_f$ is the polynomial that achieves the left hand side of Equation (18). Then we have error at most

$$\left( \frac{\pi}{2} \right)^k \frac{1}{(d+1)d\cdots(d-k+2)} \left\| f^{(k)} \right\|_\infty.$$

Using the definition of a $(k, L)$-smooth univariate function in Definition D.11 we get the error at most

$$\left( \frac{\pi}{2} \right)^k \frac{L}{(d+1)d\cdots(d-k+2)} \leq \left( \frac{\pi}{2} \right)^k \frac{L}{(d-k)^k}$$

where the inequality follows since each $d+1, d, \ldots, d-k+2 \geq (d-k)$.

Plugging in Equation (20) for $d$ we get the error is at most:

$$\left( \frac{\pi}{2} \right)^k \frac{L}{\left( \frac{\pi}{2} \right)^k \left( \sqrt[k]{\frac{L}{\epsilon}} \right)^k} = \epsilon,$$

as desired. $\qquad\qquad\square$

We will use the following definition of multivariate smooth functions that map to a single value:

**Definition D.13.** We call a $k$ times differentiable $f : [-1, 1]^n \to \mathbb{R}$ to be $(k, L)$-smooth if $\left\| \frac{\partial^k}{\partial x_m^k} f(\mathbf{x}) \right\|_\infty \leq L$ for all $1 \leq m \leq n$.

Now we show the corresponding observation for multivariate functions and polynomials:

**Corollary D.14.** *Let $\deg(P_f) = d$. For some $(k, L)$-smooth multivariate function $f$ (as in Definition D.13), then there exists a polynomial $P_f(\mathbf{x})$ with*

$$\deg(P_f) \leq O_k \left( \sqrt[k]{\frac{nL}{\epsilon}} \right)$$

*such that for all $\mathbf{x} \in [-1, 1]^n$*

$$|f(\mathbf{x}) - P_f(\mathbf{x})| \leq \epsilon.$$

*Proof.* Let $P_f$ be the polynomial we get from the left hand side of Equation (19). We want to upper bound the error as

$$\frac{c_k}{d^k} \sum_{j=1}^{n} \left\| \frac{\partial^{k+1}}{\partial x_j^{k+1}} f(\mathbf{x}) \right\|_\infty \leq \epsilon,$$

which follows if

$$\frac{c_k}{d^k} \sum_{j=1}^{n} L \leq \epsilon$$

since $f$ is $(k, L)$-smooth. The above is the same as

$$\frac{c_k n L}{d^k} \leq \epsilon,$$

or equivalently

$$\sqrt[k]{\frac{c_k n L}{\epsilon}} \leq d.$$

Picking $d = \left\lceil \sqrt[k]{\frac{c_k n L}{\epsilon}} \right\rceil$ suffices. $\qquad\square$

**Arithmetic Circuit Notation.** We briefly recall arithmetic circuits (Peter Bürgisser and Michael Clausen and M. Amin Shokrollah, 1997). An *arithmetic circuit* $\mathcal{C}$ with variables $X \triangleq \{x_1, x_2, \ldots, x_n\}$ over a field $\mathbb{F}$ is interpreted as a directed acyclic graph, where the input nodes are labelled by either the variables from $X$ or constants from $\mathbb{F}$ and the internal nodes are labelled by $+$ or $\times$ with the output being the polynomial computed at the output node.

We shall also refer to the *size*[1] of the circuit $\mathcal{C}$ as the number of wires (or edges in $\mathcal{C}$), the *depth* of the circuit as the length of the longest path between an input node and the output node, and the *width* of the circuit as the number of wires that will be intersected by a horizontal 'cut' through the circuit. Moreover, the *degree* of a circuit is defined as the degree of the polynomial computed by the circuit. We summarize this with the following definition:

**Definition D.15.** An arithmetic circuit $\mathcal{C}$ is an $(n, s, \Delta, w)$-*circuit* if $\mathcal{C}$ is an $n$-variate arithmetic circuit of size $s$, depth at most $\Delta$, and width $w$.

**BASECONV Architecture.** In the following definitions we formally define the BASECONV model (Arora et al., 2023). To formally define BASECONV, we will need the Kaleidoscope hierarchy (Dao et al., 2020) as well.

To start, we define butterfly factors:

**Definition D.16.** A **butterfly factor** of size $k \geq 2$ (denoted as $\overline{\mathbf{B}}_k$) is a matrix of the form $\overline{\mathbf{B}}_k = \begin{bmatrix} \mathbf{D}_1 & \mathbf{D}_2 \\ \mathbf{D}_3 & \mathbf{D}_4 \end{bmatrix}$ where each $\mathbf{D}_i$ is a $\frac{k}{2} \times \frac{k}{2}$ diagonal matrix. We restrict $k$ to be a power of 2.

The following definition is for a butterfly factor matrix, which is made up of the above butterfly factors:

**Definition D.17.** A **butterfly factor matrix** of size $n$ with block size $k$ (denoted as $\overline{\mathbf{B}}_k^{(n)}$) is a block diagonal matrix of $\frac{n}{k}$ (possibly different) butterfly factors of size $k$:

$$\overline{\mathbf{B}}_k^{(n)} = \mathrm{diag}\left( \left[\overline{\mathbf{B}}_k\right]_1, \left[\overline{\mathbf{B}}_k\right]_2, \ldots, \left[\overline{\mathbf{B}}_k\right]_{\frac{n}{k}} \right)$$

Now lets define a butterfly matrix:

**Definition D.18.** A **butterfly matrix** of size $n$ (denoted as $\overline{\mathbf{B}}^{(n)}$) is a matrix that can be expressed as a product of butterfly factor matrices: $\overline{\mathbf{B}}^{(n)} = \overline{\mathbf{B}}_n^{(n)} \overline{\mathbf{B}}_{\frac{n}{2}}^{(n)} \ldots \overline{\mathbf{B}}_2^{(n)}$. Equivalently, we may define $\overline{\mathbf{B}}^{(n)}$ recursively as a matrix that can be expressed in the following form:

$$\overline{\mathbf{B}}^{(n)} = \overline{\mathbf{B}}_n^{(n)} \begin{bmatrix} [\overline{\mathbf{B}}^{\left(\frac{n}{2}\right)}]_1 & 0 \\ 0 & [\overline{\mathbf{B}}^{\left(\frac{n}{2}\right)}]_2 \end{bmatrix}$$

(Note that $[\overline{\mathbf{B}}^{\left(\frac{n}{2}\right)}]_1$ and $[\overline{\mathbf{B}}^{\left(\frac{n}{2}\right)}]_2$ may be different.)

Using these butterfly matrices, lets define the Kaleidoscope Hierarchy:

---

[1]Note that if all the gates of an arithmetic circuit have bounded arity then the number of wires and gates are asymptotically the same but in this appendix we will consider gates with unbounded arity.

**Definition D.19** (The Kaleidoscope Hierarchy (Dao et al., 2020))**.**

- Define $\mathcal{B}$ as the set of all matrices that can be expressed in the form $\overline{\mathbf{B}}^{(n)}$ (for some $n$).

- Define $(\mathcal{B}\mathcal{B}^*)$ as the set of matrices $\mathbf{M}$ of the form $\mathbf{M} = \mathbf{M}_1\mathbf{M}_2^*$ for some $\mathbf{M_1}, \mathbf{M}_2 \in \mathcal{B}$.

- Define $(\mathcal{B}\mathcal{B}^*)^w$ as the set of matrices $\mathbf{M}$ that can be expressed as $\mathbf{M} = \mathbf{M}_w \ldots \mathbf{M}_2\mathbf{M}_1$, with each $\mathbf{M}_i \in (\mathcal{B}\mathcal{B}^*)\,(1 \leq i \leq w)$. (The notation $w$ represents width.)

- Define $(\mathcal{B}\mathcal{B}^*)_e^w$ as the set of $n \times n$ matrices $\mathbf{M}$ that can be expressed as $\mathbf{M} = \mathbf{S}\mathbf{E}\mathbf{S}^\top$ for some $en \times en$ matrix $\mathbf{E} \in (\mathcal{B}\mathcal{B}^*)^w$, where $\mathbf{S} \in \mathbb{F}^{n \times en} = \begin{bmatrix} \mathbf{I}_n & 0 & \ldots & 0 \end{bmatrix}]$ (i.e. $\mathbf{M}$ is the upper-left corner of $\mathbf{E}$). (The notation $e$ represents expansion relative to $n$.)

Here we now formally define a BASECONV layer:

**Definition D.20** (BASECONV (Arora et al., 2023))**.** Given an input sequence $\mathbf{u} \in \mathbb{R}^{N \times D}$, where $N$ is the sequence length and $D$ is the model dimension, a learned weight matrix $\boldsymbol{W} \in \mathbb{R}^{D \times D}$ and biases $\boldsymbol{B}_1, \boldsymbol{B}_2 \in \mathbb{R}^{N \times D}$ and a matrix of convolution filters $\boldsymbol{H} \in \mathbb{R}^{N \times D}$, a BASECONV layer computes the following:

$$\boldsymbol{y}^{\text{BASECONV}} := (\mathbf{u}\boldsymbol{W} + \boldsymbol{B}_1) \odot (\boldsymbol{H} * \mathbf{u} + \boldsymbol{B}_2) \in \mathbb{R}^{N \times D}, \tag{21}$$

where the $j$th column of $\boldsymbol{H} * \mathbf{u} \in \mathbb{R}^{N \times D}$ is defined as $\boldsymbol{H}[:, j] * \mathbf{u}[:, j]$.

The corresponding pseudocode for a BASECONV layer is as follows:

---

**Algorithm 1** BASECONV$(\mathbf{u}, \boldsymbol{W}, \boldsymbol{B}_1, \boldsymbol{H}, \boldsymbol{B}_2)$

---

**Require:** Input sequence $\mathbf{u} \in \mathbb{R}^{N \times D}$, linear map $\boldsymbol{W} \in \mathbb{R}^{D \times D}$, convolution filter $\boldsymbol{H} \in \mathbb{R}^{N \times D}$, and bias matrices $\boldsymbol{B}_1, \boldsymbol{B}_2 \in \mathbb{R}^{N \times D}$.
 1: In parallel for $0 \leq n < N : \boldsymbol{x}[n, :] = \mathbf{u}[n, :] \cdot \boldsymbol{W}$
 2: In parallel for $0 \leq t < D : \boldsymbol{z}[:, t] = \boldsymbol{H}[:, t] * \mathbf{u}[:, t]$

 3: In parallel for $0 \leq t < D : \boldsymbol{y}[:, t] \leftarrow (\boldsymbol{x}[:, t] + \boldsymbol{B}_1[:, t]) \odot (\boldsymbol{z}[:, t] + \boldsymbol{B}_2[:, t])$.       $\triangleright$ See eq. (21)
 4: **return** $\boldsymbol{y}$

---

**Remark D.21.** The definition of a BASECONV layer in Equation (22) has the input go through a linear layer before the convolution operation. For this section we will assume the linear layer is the identity matrix, as it is not needed for the results in this section.

**Assumption D.22.** Moving forward we assume the weight matrix $\boldsymbol{W} \in \mathbb{R}^{D \times D}$ in Definition D.20 also has the property $\boldsymbol{W} \in (\mathcal{B}\mathcal{B}^*)_{\text{poly-}\log D}^{\text{poly-}\log D}$. Consequently, each matrix $\boldsymbol{W}$ has $\tilde{\mathcal{O}}(D)$ parameters and runtime for matrix vector multiplication (Dao et al., 2020).

In this section, we will establish some additional basic primitives that we expect need to implement via a BASECONV layer: `shift` and `remember`. We specify them below:

**Definition D.23.** `shift`$(\boldsymbol{y}, r, t, f)$
Shift an sequential input of length $N$ up or down by $s$ entries:
INPUT: $\boldsymbol{y} \in \mathbb{R}^{N \times D}, s \geq 0$.
OUTPUT: $\boldsymbol{z} \in \mathbb{R}^{N \times D}$ where $\boldsymbol{z}^+ = $ `shift_down`$(\boldsymbol{y}, s)$ and $\boldsymbol{z}^- = $ `shift_up`$(\boldsymbol{y}, s)$

$$y \equiv \begin{pmatrix} \leftarrow y_0 \rightarrow \\ \vdots \\ \leftarrow y_{i-1} \rightarrow \\ \leftarrow y_i \rightarrow \\ \vdots \\ \leftarrow y_{N-1} \rightarrow \end{pmatrix} \qquad z^+ \equiv \begin{pmatrix} \leftarrow 0 \rightarrow \\ \vdots \\ \leftarrow 0 \rightarrow \\ \leftarrow y_0 \rightarrow \\ \vdots \\ \leftarrow y_{N-1-s} \rightarrow \end{pmatrix} \qquad z^- \equiv \begin{pmatrix} \leftarrow y_s \rightarrow \\ \vdots \\ \leftarrow y_{N-1} \rightarrow \\ \leftarrow 0 \rightarrow \\ \vdots \\ \leftarrow 0 \rightarrow \end{pmatrix}$$

The following proposition is defining the convolution Kernel that computes the `shift_down` $\left( \cdot, \lfloor \frac{N}{2} \rfloor \right)$ primitive:

**Proposition D.24.** *Define $\boldsymbol{H} \in \mathbb{R}^{2N \times D}$ as*

$$\boldsymbol{H}[k, :] = \begin{cases} \mathbf{1}^D & \text{if } k = N \\ 0 & \text{otherwise} \end{cases}.$$

*For any $\mathbf{u} \in \mathbb{R}^{2N \times D}$, $\boldsymbol{H} * \mathbf{u}$ will result in*

$$\boldsymbol{H} * \begin{pmatrix} \mathbf{u}_1 \\ \mathbf{u}_2 \end{pmatrix} \rightarrow \begin{pmatrix} \mathbf{0}^{N \times D} \\ \mathbf{u}_1 \end{pmatrix},$$

*where $\mathbf{u}_1, \mathbf{u}_2 \in \mathbb{R}^{N \times D}$.*

*Proof.* The convolution operation: $\boldsymbol{H} * \begin{pmatrix} \mathbf{u}_1 \\ \mathbf{u}_2 \end{pmatrix}$ where each column of $\boldsymbol{H}$ is convolved with each column of $\mathbf{u}$ can be restated as a polynomial multiplication. For column i, $0 \le i < 2N$,

$$\boldsymbol{H}[:, i] * \begin{pmatrix} \mathbf{u}_1 \\ \mathbf{u}_2 \end{pmatrix} [:, i] = \text{coeff}((X^N \cdot \mathbf{u}[:, i](X)) \mod X^{2N}).$$

Note that the columns of $\boldsymbol{H}$ are all $\mathbf{e}_N$ basis vectors and $\text{poly}(\mathbf{e}_N) = X^N$.

When we multiply the term through the input polynomial we get,

$$\begin{aligned} \text{coeff} &\left( X^N \cdot \left( \mathbf{u}[0][i] + \mathbf{u}[1][i]X + \cdots + \mathbf{u}[2N-1][i]X^{2N-1} \right) \mod X^{2N} \right) \\ &= \text{coeff}(\mathbf{u}[0][i]X^N + \mathbf{u}[1][i]X^{N+1} + \cdots + \mathbf{u}[2N-1][i]X^{3N-1} \mod X^{2N}). \end{aligned}$$

With the lower order terms all becoming zeros, the above is same as

$$\begin{aligned} \text{coeff}((&0 + 0X + \cdots 0X^{N-1} \\ &+ \mathbf{u}[0][i]X^N + \mathbf{u}[1][i]X^{N+1} + \cdots + \mathbf{u}[2N-1][i]X^{3N-1}) \mod X^{2N}). \end{aligned}$$

After we take the $\mod X^{2N}$ we get

$$\text{coeff}(0 + 0X + \cdots + 0X^{N-1} + \mathbf{u}[0][i]X^N + \cdots + \mathbf{u}[N-1][i]X^{2N-1}),$$

which implies that $\boldsymbol{H} * \begin{pmatrix} \mathbf{u}_1 \\ \mathbf{u}_2 \end{pmatrix}$ is

$$\begin{pmatrix} \mathbf{0}^{N \times D} \\ \mathbf{u}_1 \end{pmatrix},$$

as desired. $\qquad\square$

We also define the following primitive:

**Definition D.25.** `remember`$(\boldsymbol{y}, r, t, f)$
INPUT: $\boldsymbol{y} \in \mathbb{R}^{N' \times d'}, r \in \mathbb{Z}, t \in \mathbb{Z}, f : \mathbb{R}^{t-r} \to \mathbb{R}^{t-r+s}, \boldsymbol{v}_1 \in \mathbb{R}^r, \boldsymbol{x} \in \mathbb{R}^{t-r}$, where $\boldsymbol{y}$ is defined as below.
OUTPUT: $\boldsymbol{z} \in \mathbb{R}^{N' \times d'}$, which is defined as follows:

$$
\boldsymbol{y} \equiv \begin{pmatrix} \leftarrow \boldsymbol{v}_1 \rightarrow \\ \leftarrow \boldsymbol{x} \rightarrow \\ \boldsymbol{0}^{s \times d'} \\ \leftarrow \boldsymbol{v}_2 \rightarrow \\ \boldsymbol{0} \\ \vdots \\ \boldsymbol{0} \end{pmatrix} \qquad \boldsymbol{z} \equiv \begin{pmatrix} \leftarrow \boldsymbol{v}_1 \rightarrow \\ \leftarrow f(\boldsymbol{x}) \rightarrow \\ \leftarrow \boldsymbol{v}_2 \rightarrow \\ \boldsymbol{0} \\ \vdots \\ \boldsymbol{0} \end{pmatrix}
$$

We will need the following BASECONV implementation of `remember`:

**Proposition D.26** ((Arora et al., 2024), The Remembering Primitive). *For any $\boldsymbol{x} \in \mathbb{R}^{n \times d'}, \boldsymbol{v}_1 \in \mathbb{R}^{r \times d'}, \boldsymbol{v}_2 \in \mathbb{R}^{m-r}$ where $n = t - r$ contained in some $\boldsymbol{y} \in \mathbb{R}^{N' \times d'}$ such that $\boldsymbol{v}_1$ is in the first $r$ rows, $\boldsymbol{x}$ is in the next $n$ rows, 0s fill up the next $s$ rows, and $\boldsymbol{v}_2$ are in the next $m - r$ rows, for some $3n + 3m + 2s + 2t \leq N'$ so that for $\boldsymbol{h} \in \mathbb{R}^{n \times d}$ and $\boldsymbol{W} \in \mathbb{R}^{d' \times d'}$ with $\boldsymbol{x} * \boldsymbol{h} \in \mathbb{R}^{(n+s) \times d'}$ and $\boldsymbol{v} * \boldsymbol{h} \in \mathbb{R}^{(m+t) \times d'}$, where $\boldsymbol{v} \in \mathbb{R}^{m \times d'}$ is defined as $\boldsymbol{v}_2 + shift\_down(\boldsymbol{v}_1, m - r)$, there exists a $(N', 8, d', N', d') -$ BASECONV that computes `remember`$(\boldsymbol{y}, r, t, f)$, where $f$ can be implemented in 1 layer of BASECONV through the parameters $\boldsymbol{W} \in \mathbb{R}^{d' \times d'}, \boldsymbol{h} \in \mathbb{R}^{N' \times d'}, \boldsymbol{b}_1 \in \mathbb{R}^{N' \times d'}, \boldsymbol{b}_2 \in \mathbb{R}^{N' \times d'}$ as defined below:*

$$
f(\boldsymbol{u}) = \left( \begin{pmatrix} \boldsymbol{u}\boldsymbol{W} \\ \boldsymbol{0}^{s \times d'} \end{pmatrix} + \begin{pmatrix} \boldsymbol{b}_1 \\ \boldsymbol{1}^{s \times d'} \end{pmatrix} \right) \odot \left( \boldsymbol{u} * \boldsymbol{h} + \begin{pmatrix} \boldsymbol{b}_2 \\ \boldsymbol{0}^{s \times d'} \end{pmatrix} \right)
$$

We will also need the following generalization of the above result:

**Corollary D.27** ((Arora et al., 2023)). *Let $\boldsymbol{y}$ be as in Proposition D.26 but now let $f$ be implemented with* BASECONV$(N, L, D, N, D)$. *Then* `remember`$(\boldsymbol{y}, r, t, f)$ *where $t - r = n$ can be implemented with* BASECONV *via* $(N, O(L), D, N, D) -$ BASECONV.

The rest of Appendix D will use this $5-$tuple notation for BASECONV:

**Definition D.28.** Lets define a 5-tuple notation for a BASECONV layer as $(N, \ell, D, N', D') -$ BASECONV with $\ell$ layers such that:

1. Input and output are $N \times D$ matrices.

2. Each layer is defined by Definition D.20 where $N$ and $D$ are replaced by $N'$ and $D'$. I.e. each layer takes in $N' \times D'$ matrices and output $N' \times D'$ matrices. We refer to the tuple $(N', D')$ as the *inner dimension* of the model.

3. The matrices are projected from $(N, D) \to (N', D')$ (and vice-versa) via a linear projection.

We state the following bounds on parameters and runtime for a single BASECONV layer:

**Proposition D.29** ((Arora et al., 2023)). *An $(N, 1, D, N, D) -$ BASECONV requires $\tilde{O}(ND)$ parameters and runtime.*

We state the following result that says arithmetic circuit can be represented as a BASECONV model:

**Theorem D.30** ((Arora et al., 2023), Theorem H.21). *For any $(ND, s, \Delta, w)$-arithmetic circuit $\mathcal{C}$, there exists an equivalent $(N, \Delta', D, N', D') -$ BASECONV with $\Delta' = \mathcal{O}(\Delta \log w), N' = \mathcal{O}(w), D' = D$ that simulates $\mathcal{C}$.*

## D.2. Primitives

In this section, we provide theoretical results about primitives.

- In Appendix D.2.1, we implement the three primitives (READ, AFFINE, and MULTIPLY) from Section 3.2 using BASECONV, each using a single layer.

- Next, in Appendix D.2.2 and D.2.3, we briefly sketch how the three primitives READ, AFFINE, and MULTIPLY can be used in composition to exactly express gradient descent and Newton's method iterations on linear regression (see Appendix A).

- Finally, in Appendix D.2.4, we provide a proof that linear attention cannot exactly represent the entry-wise squaring function. As a corollary, since entry-wise square is a special case of MULTIPLY, this implies that linear attention cannot exactly express the MULTIPLY task for all arguments.

**BASECONV parameterization**   We recount the parameterization of BASECONV from Equation 2:

$$
\boldsymbol{y} := \left( \underbrace{(\boldsymbol{u} \cdot \boldsymbol{W}_{gate} + \boldsymbol{b}_{gate})}_{\textbf{Linear Projection}} \odot \underbrace{(\boldsymbol{h} * (\boldsymbol{u} \cdot \boldsymbol{W}_{in} + \boldsymbol{b}_{in}) + \boldsymbol{b}_{conv})}_{\textbf{Convolution}} \right) \cdot \boldsymbol{W}_{out} + \boldsymbol{b}_{out}
$$

$$
:= W_{out}(W_{gate}(\mathbf{u}) \odot Conv(W_{in}(\mathbf{u})))
$$

(22)

where $W_{in}, W_{gate}, W_{out}$ are linear projections $\mathbb{R}^D \to \mathbb{R}^D$.

### D.2.1. 1-LAYER BASECONV CAN IMPLEMENT LINEAR ALGEBRA PRIMITIVES

Below, we recount the definitions of our linear algebra primitives from Section 3.2 and describe our BASECONV weight constructions.

**Read**   The READ operator is:

$$
\text{READ}(i, j, a, b)(\mathbf{u}) = \begin{cases} \mathbf{u}[k, a:b] & k \neq j \\ \mathbf{u}[i, a:b] & k = j \end{cases}.
$$

(23)

Our implementation requires the use of the positional encodings and residual connections within the BASECONV architecture. Concretely, consider the input

$$
\boldsymbol{u}_{in} = \left( \begin{array}{cccc} \boldsymbol{e}_1 & \boldsymbol{e}_2 & \dots & \boldsymbol{e}_N \\ \hline \boldsymbol{u}[1,:] & \boldsymbol{u}[2,:] & \dots & \boldsymbol{u}[N,:] \end{array} \right),
$$

where the basis vector $\boldsymbol{e}_k$ represents the positional encoding for the $k$-th entry of the sequence. Define the output of the BASECONV layer *with residual connection*:

$$
\boldsymbol{y} := W_{out}(W_{gate}(\mathbf{u}) \odot Conv(W_{in}(\mathbf{u})) + \boldsymbol{u}).
$$

Then the following weight construction is equivalent to READ$(i, j, a, b)$:

- $W_{gate}(\boldsymbol{u}[k,:]) := \boldsymbol{u}[k, j] \mathbf{1}^D$

- $Conv(W_{in}(\boldsymbol{u}))[k,:] := \boldsymbol{u}[k+i-j,:] - \boldsymbol{u}[k,:]$

- $W_{out} := proj(a:b)$.

In particular, $W_{gate}$ is defined such that

$$
W_{gate}(\boldsymbol{u}[k,:]) = \begin{cases} \mathbf{1}^D & k = j \\ \mathbf{0}^D & k \neq j \end{cases}.
$$

Thus

$$W_{gate}(\mathbf{u}) \odot Conv(W_{in}(\mathbf{u})) = \begin{cases} \mathbf{u}[k+i-j,:] - \mathbf{u}[k,:] = \mathbf{u}[i,:] - \mathbf{u}[j,:] & k = j \\ \mathbf{0}^D & k \neq j \end{cases}.$$

Finally,

$$W_{gate}(\mathbf{u}) \odot Conv(W_{in}(\mathbf{u})) + \mathbf{u} = \begin{cases} \mathbf{u}[i,:] & k = j \\ \mathbf{u}[k,:] & k \neq j \end{cases}$$

so the final output of this layer will be exactly equivalent to $\text{READ}(i, j, a, b)$.

**Affine transformation** The AFFINE operator is:

$$\text{AFFINE}(\boldsymbol{H})(\mathbf{u}) = \mathbf{u}\boldsymbol{H} \tag{24}$$

Define $Conv(W_{in}(\mathbf{u})) = \mathbf{1}_D$, $W_{gate} = I$, and $W_{out} = \boldsymbol{H}$. Then

$$W_{gate}(\mathbf{u}) \odot Conv(W_{in}(\mathbf{u})) = \boldsymbol{u}$$

so

$$W_{out}(W_{gate}(\mathbf{u}) \odot Conv(W_{in}(\mathbf{u}))) = \boldsymbol{u}\boldsymbol{H}.$$

Thus the output of this layer is exactly equivalent to $\text{AFFINE}(\boldsymbol{H})$.

**Element-wise multiply** The MULTIPLY operator is:

$$\text{MULTIPLY}(a, b, d_{out})(\mathbf{u}) = \mathbf{u}[:, a : a + d_{out}] \odot \mathbf{u}[:, b : b + d_{out}] \tag{25}$$

Define $Conv = \text{Identity}$, $W_{in} = proj(a : a + d_{out})$, $W_{gate} = proj(b : b + d_{out})$, and $W_{out} = \boldsymbol{I}$.

Then

$$W_{gate}(\mathbf{u}) \odot Conv(W_{in}(\mathbf{u})) = \mathbf{u}[:, a : a + d_{out}] \odot \mathbf{u}[:, b : b + d_{out}].$$

Since $W_{out} = \boldsymbol{I}$, the output of this layer will be equivalent to $\text{MULTIPLY}(a, b, d_{out})$.

### D.2.2. GRADIENT DESCENT

We assume our input is of the form

$$\boldsymbol{u} = \begin{pmatrix} \boldsymbol{x}_1 & \dots & \boldsymbol{x}_N & \boldsymbol{w}_0 \\ y_1 & \dots & y_N & 0 \end{pmatrix}.$$

Our goal is to compute the gradient update

$$\boldsymbol{w}_1 := \boldsymbol{w}_0 - \frac{\eta}{N} \sum_{i=1}^{N} (\boldsymbol{w}_0^T \boldsymbol{x}_i - y_i) \boldsymbol{x}_i. \tag{26}$$

Intuitively, our argument proceeds similarly to the causal gradient descent construction from Appendix D.3.1:

- First, we repeatedly apply READ and AFFINE to move the information $\{\boldsymbol{x}_i, y_i\} \forall i$ into e.g. the final entry of the sequence. Without loss of generality, we omit the rest of the sequence, and assume we have access to a large enough embedding dimension that we can make use of arbitrary amounts of memory.

  After this phase, our $\boldsymbol{u}$ is of the form

$$\dots \begin{pmatrix} \boldsymbol{w}_0 & 0 & \boldsymbol{x}_1 & \dots & \boldsymbol{x}_N & y_1 & \dots & y_N & \dots \end{pmatrix}^T.$$

- Next, we use MULTIPLY and AFFINE to compute and store $\{\boldsymbol{w}_0^T \boldsymbol{x}_i\}$ for all $i$. We will end up with

$$\boldsymbol{u} = \dots \begin{pmatrix} \boldsymbol{w}_0 & 0 & \{\boldsymbol{x}_i\}_i & \{y_i\}_i & \{\boldsymbol{w}_0^T \boldsymbol{x}_i\}_i & \dots \end{pmatrix}.$$

- We use AFFINE to compute and store $\{\boldsymbol{w}_0^T \boldsymbol{x}_i - y_i\}$ for all $i$:

$$\boldsymbol{u} = \ldots \begin{pmatrix} \boldsymbol{w}_0 & 0 & \{\boldsymbol{x}_i\}_i & \{y_i\}_i & \{\boldsymbol{w}_0^T \boldsymbol{x}_i\}_i & \{\boldsymbol{w}_0^T \boldsymbol{x}_i - y_i\}_i & \ldots \end{pmatrix}.$$

- We use MULTIPLY and AFFINE to compute and store $\{(\boldsymbol{w}_0^T \boldsymbol{x}_i - y_i)\boldsymbol{x}_i\}$ for all $i$:

$$\boldsymbol{u} = \ldots \begin{pmatrix} \boldsymbol{w}_0 & 0 & \{\boldsymbol{x}_i\}_i & \{y_i\}_i & \{\boldsymbol{w}_0^T \boldsymbol{x}_i\}_i & \{(\boldsymbol{w}_0^T \boldsymbol{x}_i - y_i)\boldsymbol{x}_i\}_i & \ldots \end{pmatrix}.$$

- Finally, we can use AFFINE to compute the gradient update:

$$\boldsymbol{u} = \ldots \begin{pmatrix} \boldsymbol{w}_0 - \frac{\eta}{N}\sum_{i=1}^N(\boldsymbol{w}_0^T \boldsymbol{x}_i - y_i)\boldsymbol{x}_i & 0 & \{\boldsymbol{x}_i\}_i & \{y_i\}_i & \{\boldsymbol{w}_0^T \boldsymbol{x}_i\}_i & \{(\boldsymbol{w}_0^T \boldsymbol{x}_i - y_i)\boldsymbol{x}_i\}_i & \ldots \end{pmatrix}.$$

### D.2.3. NEWTON'S METHOD

We assume our input is of the form

$$\boldsymbol{u} = \begin{pmatrix} \boldsymbol{x}_1 & \ldots & \boldsymbol{x}_N & \boldsymbol{A}_0[1,:] & \ldots & \boldsymbol{A}_0[D,:] \\ y_1 & \ldots & y_N & 0 & \ldots & 0 \end{pmatrix}.$$

Our goal is to compute the Newton's iterate:

$$\boldsymbol{A}_1 := \boldsymbol{A}_0(2\boldsymbol{I} - (\boldsymbol{X}^T\boldsymbol{X})\boldsymbol{A}_0), \tag{27}$$

where

$$\boldsymbol{X} = \begin{pmatrix} \leftarrow \boldsymbol{x}_1 \rightarrow \\ \vdots \\ \leftarrow \boldsymbol{x}_N \rightarrow \end{pmatrix}, \quad \boldsymbol{y} = \begin{pmatrix} y_1 \\ \vdots \\ y_N \end{pmatrix}. \tag{28}$$

For any matrix $\boldsymbol{M} \in \mathbb{R}^{n \times p}$, let $flt$ denote the `flatten` operation, so that $flt(\boldsymbol{M})$ represent a vectorized version of $\boldsymbol{M}$: $flt(\boldsymbol{M}) \in \mathbb{R}^{np}$.

We proceed similarly to the argument from Appendix D.2.2.

- First, we repeatedly apply READ and AFFINE to move all information $\{\boldsymbol{x}_i\}_i \,\forall i$ and $flt(\boldsymbol{A})$ to e.g. the final entry of the sequence. We omit the rest of the sequence for notational ease, and we assume we have access to a large enough embedding dimension that we can make use of arbitrary amounts of memory.

  After this phase, we have
$$\boldsymbol{u} = \ldots \begin{pmatrix} flt(\boldsymbol{A}_0) & \{\boldsymbol{x}_i\}_i & \ldots \end{pmatrix}.$$

- Using AFFINE, we can copy and rearrange the $\boldsymbol{x}_i$'s to construct copies of $flt(\boldsymbol{X})$ and $flt(\boldsymbol{X}^T)$:

$$\boldsymbol{u} = \ldots \begin{pmatrix} flt(\boldsymbol{A}_0) & \{\boldsymbol{x}_i\}_i & flt(\boldsymbol{X}^T) & flt(\boldsymbol{X}) & \ldots \end{pmatrix}.$$

- Now, note that we can represent the matrix multiplication $\boldsymbol{X}^T\boldsymbol{X}$ as a linear combination of the entries of the element-wise multiplication $flt(\boldsymbol{X}^T) \odot flt(\boldsymbol{X})$. This means that we can obtain $flt(\boldsymbol{X}^T\boldsymbol{X})$ using a single application of MULTIPLY and AFFINE:

$$\boldsymbol{u} = \ldots \begin{pmatrix} flt(\boldsymbol{A}_0) & \{\boldsymbol{x}_i\}_i & flt(\boldsymbol{X}^T) & flt(\boldsymbol{X}) & flt(\boldsymbol{X}^T\boldsymbol{X}) \ldots \end{pmatrix}.$$

- By the same argument, we can obtain $flt((\boldsymbol{X}^T\boldsymbol{X})\boldsymbol{A}_0)$ using another application of MULTIPLY and AFFINE:

$$\boldsymbol{u} = \ldots \begin{pmatrix} flt(\boldsymbol{A}_0) & \{\boldsymbol{x}_i\}_i & flt(\boldsymbol{X}^T) & flt(\boldsymbol{X}) & flt((\boldsymbol{X}^T\boldsymbol{X})\boldsymbol{A}_0) \ldots \end{pmatrix}.$$

- Finally, we have that $flt(\boldsymbol{A}_1) := 2flt(\boldsymbol{A}_0) - flt((\boldsymbol{X}^T\boldsymbol{X})\boldsymbol{A}_0)$ can be obtained using AFFINE once more:

$$\boldsymbol{u} = \ldots \begin{pmatrix} flt(\boldsymbol{A}_1) & \{\boldsymbol{x}_i\}_i & flt(\boldsymbol{X}^T) & flt(\boldsymbol{X}) & flt((\boldsymbol{X}^T\boldsymbol{X})\boldsymbol{A}_0) \ldots \end{pmatrix}.$$

D.2.4. ATTENTION CAN'T IMPLEMENT ELEMENT-WISE SQUARING.

In this section, we consider the following parameterization of *linear attention*:

$$\text{LinearAttn}(\boldsymbol{u}) = (\boldsymbol{u}\boldsymbol{W_Q})(\boldsymbol{u}\boldsymbol{W_K})^T(\boldsymbol{u}\boldsymbol{W_V} + \boldsymbol{B}), \tag{29}$$

where $\boldsymbol{u} \in \mathbb{R}^{N \times D}, \boldsymbol{W_Q}, \boldsymbol{W_K}, \boldsymbol{W_V}, \boldsymbol{B} \in \mathbb{R}^{D \times D}$.

**Theorem D.31.** *One-layer linear attention cannot exactly represent the entry-wise squaring function* SQUARE $: \mathbb{R}^{N \times D} \to \mathbb{R}^{N \times D}$ *s.t.*

$$\text{SQUARE}(\boldsymbol{u})_{ij} = \boldsymbol{u}_{ij}^2$$

*for all* $\boldsymbol{u} \in \mathbb{R}^{N \times D}$.

*Proof.* We proceed by contradiction. Let's assume there exists $\boldsymbol{W_Q}, \boldsymbol{W_K}, \boldsymbol{W_V}, \boldsymbol{B} \in \mathbb{R}^{D \times D}$ such that $\forall \boldsymbol{u} \in \mathbb{R}^{N \times D}$,

$$(\boldsymbol{u}\boldsymbol{W_Q})(\boldsymbol{u}\boldsymbol{W_K})^T(\boldsymbol{u}\boldsymbol{W_V} + \boldsymbol{B}) = \text{SQUARE}(\boldsymbol{u}). \tag{30}$$

Consider the set of inputs $\boldsymbol{u} \in \mathbb{R}^{N \times D}$ with two non-zero entries, defined as

$$\boldsymbol{u}_{ij} = \begin{cases} \boldsymbol{u}_{ij} & (i,j) \in \{(a,c), (b,d)\} \\ 0 & \text{else} \end{cases} \tag{31}$$

for an arbitrary choice of $a, b \in [N], c, d \in [D]$. Then:

$$\boldsymbol{Q} := \boldsymbol{u}\boldsymbol{W_Q} = \begin{pmatrix} \boldsymbol{0}^N \\ \vdots \\ \boldsymbol{0}^N \\ \boldsymbol{u}_{ac}\boldsymbol{W_Q}[c, :] \\ \boldsymbol{0}^N \\ \vdots \\ \boldsymbol{0}^N \\ \boldsymbol{u}_{bd}\boldsymbol{W_Q}[d, :] \\ \boldsymbol{0}^N \\ \vdots \\ \boldsymbol{0}^N \end{pmatrix} \tag{32}$$

where $\boldsymbol{Q}$'s rows are all $\boldsymbol{0}$ except for the $a$-th and $b$-th, which are $\boldsymbol{u}_{ac}\boldsymbol{W_Q}[c, :]$ and $\boldsymbol{u}_{bd}\boldsymbol{W_Q}[d, :]$ respectively.

Similarly:

$$
\boldsymbol{K} := \boldsymbol{u}\boldsymbol{W_K} =
\begin{pmatrix}
\mathbf{0}^N \\
\vdots \\
\mathbf{0}^N \\
\boldsymbol{u}_{ac}\boldsymbol{W_K}[c,:] \\
\mathbf{0}^N \\
\vdots \\
\mathbf{0}^N \\
\boldsymbol{u}_{bd}\boldsymbol{W_K}[d,:] \\
\mathbf{0}^N \\
\vdots \\
\mathbf{0}^N
\end{pmatrix}
\tag{33}
$$

and

$$
\boldsymbol{V} := \boldsymbol{u}\boldsymbol{W_V} =
\begin{pmatrix}
\mathbf{0}^N \\
\vdots \\
\mathbf{0}^N \\
\boldsymbol{u}_{ac}\boldsymbol{W_V}[c,:] \\
\mathbf{0}^N \\
\vdots \\
\mathbf{0}^N \\
\boldsymbol{u}_{bd}\boldsymbol{W_V}[d,:] \\
\mathbf{0}^N \\
\vdots \\
\mathbf{0}^N
\end{pmatrix}
\tag{34}
$$

Then the attention matrix, $\boldsymbol{A} = \boldsymbol{Q}\boldsymbol{K}^T$, satisfies

$$\boldsymbol{A}_{ij} = \begin{cases} \boldsymbol{u}_{ac}^2 (\boldsymbol{W_Q}\boldsymbol{W_K}^T)_{cc} & (i,j) = (a,a) \\ \boldsymbol{u}_{ac}\boldsymbol{u}_{bd}(\boldsymbol{W_Q}\boldsymbol{W_K}^T)_{cd} & (i,j) = (a,b) \\ \boldsymbol{u}_{ac}\boldsymbol{u}_{bd}(\boldsymbol{W_Q}\boldsymbol{W_K}^T)_{dc} & (i,j) = (b,a) \\ \boldsymbol{u}_{bd}^2 (\boldsymbol{W_Q}\boldsymbol{W_K}^T)_{dd} & (i,j) = (b,b) \\ 0 & \text{else} \end{cases} . \tag{35}$$

Now let's consider the output of linear attention:

$$\boldsymbol{O} = (\boldsymbol{Q}\boldsymbol{K}^T)(\boldsymbol{V} + \boldsymbol{B}) \tag{36}$$

such that $\boldsymbol{O} = \text{SQUARE}(\boldsymbol{u})$.

**Case 1: $\boldsymbol{B} = 0$.** We have

$$\boldsymbol{O}[a,:] = \boldsymbol{u}_{ac}^3 (\boldsymbol{W_Q}\boldsymbol{W_K}^T)_{cc}\boldsymbol{W_V}[c,:] + \boldsymbol{u}_{ac}\boldsymbol{u}_{bd}^2 (\boldsymbol{W_Q}\boldsymbol{W_K}^T)_{cd}\boldsymbol{W_V}[d,:] \tag{37}$$

and

$$\boldsymbol{O}[b,:] = \boldsymbol{u}_{ac}^2 \boldsymbol{u}_{bd}(\boldsymbol{W_Q}\boldsymbol{W_K}^T)_{dc}\boldsymbol{W_V}[c,:] + \boldsymbol{u}_{bd}^3 (\boldsymbol{W_Q}\boldsymbol{W_K}^T)_{dd}\boldsymbol{W_V}[d,:] \tag{38}$$

Note that each term of the output is a cubic polynomial of the inputs $\boldsymbol{u}_{ac}$ and $\boldsymbol{u}_{bd}$, whereas our target $\text{SQUARE}(\boldsymbol{u})$ consists of quadratic polynomials, so these cannot be exactly equivalent.

**Case 2: $\boldsymbol{B} \neq 0$.** In this case,

$$\boldsymbol{O}[a,:] = \boldsymbol{u}_{ac}^3 (\boldsymbol{W_Q}\boldsymbol{W_K}^T)_{cc}\boldsymbol{W_V}[c,:] + \boldsymbol{u}_{ac}^2 (\boldsymbol{W_Q}\boldsymbol{W_K}^T)_{cc}\boldsymbol{B}[a,:] + \boldsymbol{u}_{ac}\boldsymbol{u}_{bd}^2 (\boldsymbol{W_Q}\boldsymbol{W_K}^T)_{cd}\boldsymbol{W_V}[d,:] + \boldsymbol{u}_{ac}\boldsymbol{u}_{bd}(\boldsymbol{W_Q}\boldsymbol{W_K}^T)_{cd}\boldsymbol{B}[b,:] \tag{39}$$

and

$$\boldsymbol{O}[b,:] = \boldsymbol{u}_{ac}^2 \boldsymbol{u}_{bd}(\boldsymbol{W_Q}\boldsymbol{W_K}^T)_{dc}\boldsymbol{W_V}[c,:] + \boldsymbol{u}_{ac}\boldsymbol{u}_{bd}(\boldsymbol{W_Q}\boldsymbol{W_K}^T)_{dc}\boldsymbol{B}[a,:] + \boldsymbol{u}_{bd}^3 (\boldsymbol{W_Q}\boldsymbol{W_K}^T)_{dd}\boldsymbol{W_V}[d,:] + \boldsymbol{u}_{bd}^2 (\boldsymbol{W_Q}\boldsymbol{W_K})_{dd}\boldsymbol{B}[b,:] \tag{40}$$

In order for $\boldsymbol{O} = \text{SQUARE}(\boldsymbol{u})$, we need

$$\boldsymbol{O}[a,:] = \boldsymbol{u}_{ac}^2 \boldsymbol{e}_c^D, \quad \boldsymbol{O}[b,:] = \boldsymbol{u}_{bd}^2 \boldsymbol{e}_d^D \tag{41}$$

Then, setting the quadratic terms of Equation 41 and Equations 39, 40 equal, we must have

$$(\boldsymbol{W_Q}\boldsymbol{W_K}^T)_{cc} = (\boldsymbol{W_Q}\boldsymbol{W_K}^T)_{dd} = 1 \tag{42}$$

and

$$\boldsymbol{B}[a,:] = \boldsymbol{e}_a^D, \quad \boldsymbol{B}[b,:] = \boldsymbol{e}_b^D \tag{43}$$

The cubic terms in Equations 39, 40 must also vanish, which implies

$$\boldsymbol{W_V}[c,:] = \boldsymbol{W_V}[d,:] = \boldsymbol{0}^D. \tag{44}$$

The $\boldsymbol{u}_{ac}\boldsymbol{u}_{bd}$ terms must also vanish, which implies

$$(\boldsymbol{W_Q}\boldsymbol{W_K}^T)_{cd} = (\boldsymbol{W_Q}\boldsymbol{W_K}^T)_{dc} = 0. \tag{45}$$

Finally, note that the above must hold for all choices of $a, b \in [N]$ and $c, d \in [D]$. This implies that we have:

$$\boldsymbol{V} = \boldsymbol{0}^{D \times D}, \quad , \boldsymbol{B} = \boldsymbol{I}^{D \times D}, \quad \boldsymbol{W_Q}\boldsymbol{W_K}^T = \boldsymbol{I}^{D \times D} \tag{46}$$

In other words, the set of constraints from our arguments above fully specify the weights of linear attention. However, we can verify that these weights fail to express SQUARE by evaluating the linear attention:

$$(\boldsymbol{W_Q}\boldsymbol{W_K}^T)(\boldsymbol{V} + \boldsymbol{B}) = (\boldsymbol{u}\boldsymbol{W_Q}\boldsymbol{W_K}^T\boldsymbol{u}^T)(\boldsymbol{u}\boldsymbol{V} + \boldsymbol{B}) = (\boldsymbol{u}\boldsymbol{u}^T)(\boldsymbol{0}^{D \times D} + \boldsymbol{I}^{D \times D}) = \boldsymbol{u}\boldsymbol{u}^T \tag{47}$$

However, it is easy to check that $\boldsymbol{u}\boldsymbol{u}^T \neq \text{SQUARE}(\boldsymbol{u})$, which completes the proof by contradiction.

## D.3. Upper and lower bounds with BASECONV for gradient descent

In this section, we detail upper and lower bounds for implementing gradient descent using BASECONV, as discussed in Section 4.

- **Upper bounds.** We provide two explicit constructions for implementing iterations gradient descent on linear regression: one for *non-causal* BASECONV requiring $O(1)$ layers and $O(D)$ state size, and one for *causal* BASECONV requiring $O(1)$ layers and $O(D^2)$ state size.

- **Lower bounds.** In Appendix D.3.2, we prove that our constructions are asymptotically optimal with respect to layers and state size.

### D.3.1. UPPER BOUNDS: BASECONV CAN IMPLEMENT GRADIENT DESCENT FOR LINEAR REGRESSION

In this section, we provide weight constructions for exactly implementing gradient descent on linear regression. Recall:

$$\mathcal{L}_N = \frac{1}{2N} \sum_{i=1}^{N} (\boldsymbol{w}^T \boldsymbol{x}_i - \boldsymbol{y}_i)^2 \tag{48}$$

so

$$\nabla_{\boldsymbol{w}} \mathcal{L}_N = \frac{1}{N} \sum_{i=1}^{N} (\boldsymbol{w}^T \boldsymbol{x}_i - \boldsymbol{y}_i) \boldsymbol{x}_i \tag{49}$$

$$= \frac{1}{N} \left( \sum_{i=1}^{N} \boldsymbol{y}_i \boldsymbol{x}_i - \left( \sum_{i=1}^{N} \boldsymbol{x}_i \boldsymbol{x}_i^T \right) \boldsymbol{w} \right) \tag{50}$$

**Non-causal BASECONV** This weight construction uses Equation 49 to compute the gradient descent update.

We note that non-causal constructions for in-context linear regression are standard in the literature: e.g. (Von Oswald et al., 2023; Ahn et al., 2024).

We start with input:

$$\boldsymbol{y} \equiv \begin{pmatrix} \boldsymbol{x}_1 & \dots & \boldsymbol{x}_N & \boldsymbol{x}_q \\ \boldsymbol{y}_1 & \dots & \boldsymbol{y}_N & 0 \end{pmatrix}$$

We define the initial embedding:

$$\begin{pmatrix} \boldsymbol{x}_1 & \dots & \boldsymbol{x}_N & \boldsymbol{0}^D \\ \boldsymbol{y}_1 & \dots & \boldsymbol{y}_N & 0 \\ \boldsymbol{w}_0 & \dots & \boldsymbol{w}_0 & \boldsymbol{w}_0 \\ \boldsymbol{0}^D & \dots & \boldsymbol{0}^D & \boldsymbol{0}^D \\ \boldsymbol{0}^D & \dots & \boldsymbol{0}^D & \boldsymbol{0}^D \\ \boldsymbol{0}^D & \dots & \boldsymbol{0}^D & \boldsymbol{x}_q \\ 0 & \dots & 0 & 0 \end{pmatrix}$$

We drop the bottom two rows of the block matrix representation for now and show how to perform the gradient descent update with the rest of the embedding.

Layer 1:

$$
\underbrace{\begin{pmatrix} \leftarrow \boldsymbol{x}_i \rightarrow \\ \leftarrow \boldsymbol{y}_i \rightarrow \\ \leftarrow \boldsymbol{w}_0 \rightarrow \\ \leftarrow \boldsymbol{x}_i \rightarrow \\ \leftarrow \boldsymbol{0}^D \rightarrow \end{pmatrix}}_{conv(in\_proj(\cdot))} \odot \underbrace{\begin{pmatrix} \leftarrow \boldsymbol{1}^D \rightarrow \\ \leftarrow \boldsymbol{1} \rightarrow \\ \leftarrow \boldsymbol{1}^D \rightarrow \\ \leftarrow \boldsymbol{w}_0 \rightarrow \\ \leftarrow \boldsymbol{0}^D \rightarrow \end{pmatrix}}_{gate\_proj(\cdot)} = \begin{pmatrix} \leftarrow \boldsymbol{x}_i \rightarrow \\ \leftarrow \boldsymbol{y}_i \rightarrow \\ \leftarrow \boldsymbol{w}_0 \rightarrow \\ \leftarrow \boldsymbol{x}_i \odot \boldsymbol{w}_0 \rightarrow \\ \leftarrow \boldsymbol{0}^D \rightarrow \end{pmatrix}
$$

$$
\begin{pmatrix} \leftarrow \boldsymbol{x}_i \rightarrow \\ \leftarrow \boldsymbol{y}_i \rightarrow \\ \leftarrow \boldsymbol{w}_0 \rightarrow \\ \leftarrow \boldsymbol{x}_i \odot \boldsymbol{w}_0 \rightarrow \\ \leftarrow \boldsymbol{0}^D \rightarrow \end{pmatrix} \underbrace{\rightarrow}_{out\_proj(\cdot)} \begin{pmatrix} \leftarrow \boldsymbol{x}_i \rightarrow \\ \leftarrow \boldsymbol{y}_i \rightarrow \\ \leftarrow \boldsymbol{w}_0 \rightarrow \\ \leftarrow \boldsymbol{x}_i \odot \boldsymbol{w}_0 \rightarrow \\ \leftarrow (\boldsymbol{w}_0^T \boldsymbol{x}_i - \boldsymbol{y}_i)\boldsymbol{1}^D \rightarrow \end{pmatrix}
$$

Layer 2:

$$
\underbrace{\begin{pmatrix} \leftarrow \boldsymbol{x}_i \rightarrow \\ \leftarrow \boldsymbol{y}_i \rightarrow \\ \leftarrow \boldsymbol{w}_0 \rightarrow \\ \leftarrow \boldsymbol{x}_i \odot \boldsymbol{w}_0 \rightarrow \\ \leftarrow (\boldsymbol{w}_0^T \boldsymbol{x}_i - \boldsymbol{y}_i)\boldsymbol{1}^D \rightarrow \end{pmatrix}}_{conv(in\_proj(\cdot))} \odot \underbrace{\begin{pmatrix} \leftarrow \boldsymbol{1}^D \rightarrow \\ \leftarrow \boldsymbol{1} \rightarrow \\ \leftarrow \boldsymbol{1}^D \rightarrow \\ \leftarrow \boldsymbol{1}^D \rightarrow \\ \leftarrow \boldsymbol{x}_i \rightarrow \end{pmatrix}}_{gate\_proj(\cdot)} = \begin{pmatrix} \leftarrow \boldsymbol{x}_i \rightarrow \\ \leftarrow \boldsymbol{y}_i \rightarrow \\ \leftarrow \boldsymbol{w}_0 \rightarrow \\ \leftarrow \boldsymbol{x}_i \odot \boldsymbol{w}_0 \rightarrow \\ \leftarrow (\boldsymbol{w}_0^T \boldsymbol{x}_i - \boldsymbol{y}_i)\boldsymbol{x}_i \rightarrow \end{pmatrix}
$$

$$
\begin{pmatrix} \leftarrow \boldsymbol{x}_i \rightarrow \\ \leftarrow \boldsymbol{y}_i \rightarrow \\ \leftarrow \boldsymbol{w}_0 \rightarrow \\ \leftarrow \boldsymbol{x}_i \odot \boldsymbol{w}_0 \rightarrow \\ \leftarrow (\boldsymbol{w}_0^T \boldsymbol{x}_i - \boldsymbol{y}_i)\boldsymbol{x}_i \rightarrow \end{pmatrix} \underbrace{\rightarrow}_{out\_proj(\cdot)=Identity} \begin{pmatrix} \leftarrow \boldsymbol{x}_i \rightarrow \\ \leftarrow \boldsymbol{y}_i \rightarrow \\ \leftarrow \boldsymbol{w}_0 \rightarrow \\ \leftarrow \boldsymbol{x}_i \odot \boldsymbol{w}_0 \rightarrow \\ \leftarrow (\boldsymbol{w}_0^T \boldsymbol{x}_i - \boldsymbol{y}_i)\boldsymbol{x}_i \rightarrow \end{pmatrix}
$$

Layer 3:

$$
\begin{pmatrix} \leftarrow \boldsymbol{x}_i \rightarrow \\ \leftarrow \boldsymbol{y}_i \rightarrow \\ \leftarrow \boldsymbol{w}_0 \rightarrow \\ \leftarrow \boldsymbol{x}_i \odot \boldsymbol{w}_0 \rightarrow \\ \leftarrow (\boldsymbol{w}_0^T \boldsymbol{x}_i - \boldsymbol{y}_i)\boldsymbol{x}_i \rightarrow \end{pmatrix} \underbrace{\rightarrow}_{conv(in\_proj(\cdot))} \begin{pmatrix} \leftarrow \boldsymbol{x}_i \rightarrow \\ \leftarrow \boldsymbol{y}_i \rightarrow \\ \leftarrow \boldsymbol{w}_0 \rightarrow \\ \leftarrow \boldsymbol{x}_i \odot \boldsymbol{w}_0 \rightarrow \\ \leftarrow \sum_{i=1}^N (\boldsymbol{w}_0^T \boldsymbol{x}_i - \boldsymbol{y}_i)\boldsymbol{x}_i \rightarrow \end{pmatrix}
$$

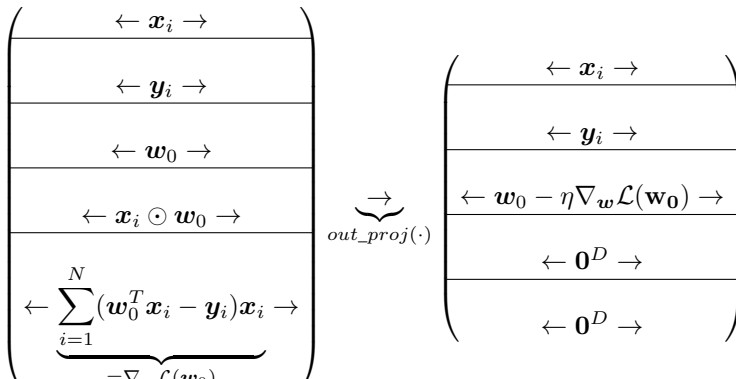

After performing arbitrarily many gradient updates, a final BASECONV layer can be used to compute $\hat{\boldsymbol{w}}^T \boldsymbol{x}_q$.

**Causal BASECONV** This weight construction uses Equation 50 to compute the gradient descent update.

We start with input:

$$
\boldsymbol{y} \equiv
\begin{pmatrix}
\boldsymbol{x}_1 & \ldots & \boldsymbol{x}_N & \boldsymbol{0}^D \\
\boldsymbol{y}_1 & \ldots & \boldsymbol{y}_N & 0 \\
\boldsymbol{0}^D & \ldots & \boldsymbol{0}^D & \boldsymbol{w}_0
\end{pmatrix}
$$

We use two BASECONV layers to construct an initial embedding, after which each gradient descent update step will only require a single BASECONV layer.

In the following construction, we use $flt$ to denote the `flatten` operation, which maps an $M \times N$ matrix to a $MN$-entry vector with the same elements.

Layer 1:

$$
\underbrace{
\begin{pmatrix}
\boldsymbol{x}_1 & \ldots & \boldsymbol{x}_N & \boldsymbol{0}^D \\
\boldsymbol{y}_1 & \ldots & \boldsymbol{y}_N & 0 \\
\boldsymbol{0}^D & \ldots & \boldsymbol{0}^D & \boldsymbol{w}_0 \\
\boldsymbol{x}_1 & \ldots & \boldsymbol{x}_N & \boldsymbol{0}^D \\
flt(\boldsymbol{x}_1(\boldsymbol{1}^D)^T) & \ldots & flt(\boldsymbol{x}_N(\boldsymbol{1}^D)^T) & flt(\boldsymbol{0}^D(\boldsymbol{0}^D)^T)
\end{pmatrix}
}_{conv(in\_proj(\cdot))}
\odot
\underbrace{
\begin{pmatrix}
\leftarrow \boldsymbol{1}^D \rightarrow \\
\leftarrow \boldsymbol{1} \rightarrow \\
\leftarrow \boldsymbol{1}^D \rightarrow \\
\boldsymbol{y}_1\boldsymbol{1}^D \quad \ldots \quad \boldsymbol{y}_N\boldsymbol{1}^D \quad \boldsymbol{0}^D \\
flt(\boldsymbol{1}^D\boldsymbol{x}_1^T) \quad \ldots \quad flt(\boldsymbol{1}^D\boldsymbol{x}_N^T) \quad flt(\boldsymbol{0}^D(\boldsymbol{0}^D)^T)
\end{pmatrix}
}_{gate\_proj(\cdot)}
=
$$

$$
\begin{pmatrix}
\boldsymbol{x}_1 & \ldots & \boldsymbol{x}_N & \boldsymbol{0}^D \\
\boldsymbol{y}_1 & \ldots & \boldsymbol{y}_N & 0 \\
\boldsymbol{0}^D & \ldots & \boldsymbol{0}^D & \boldsymbol{w}_0 \\
\boldsymbol{y}_1\boldsymbol{x}_1 & \ldots & \boldsymbol{y}_1\boldsymbol{x}_N & \boldsymbol{0}^D \\
flt(\boldsymbol{x}_1\boldsymbol{x}_1^T) & \ldots & flt(\boldsymbol{x}_N\boldsymbol{x}_N^T) & flt(\boldsymbol{0}^D(\boldsymbol{0}^D)^T)
\end{pmatrix}
\underset{out\_proj=Identity}{\longrightarrow}
\begin{pmatrix}
\boldsymbol{x}_1 & \ldots & \boldsymbol{x}_N & \boldsymbol{0}^D \\
\boldsymbol{y}_1 & \ldots & \boldsymbol{y}_N & 0 \\
\boldsymbol{0}^D & \ldots & \boldsymbol{0}^D & \boldsymbol{w}_0 \\
\boldsymbol{y}_1\boldsymbol{x}_1 & \ldots & \boldsymbol{y}_1\boldsymbol{x}_N & \boldsymbol{0}^D \\
flt(\boldsymbol{x}_1\boldsymbol{x}_1^T) & \ldots & flt(\boldsymbol{x}_N\boldsymbol{x}_N^T) & flt(\boldsymbol{0}^D(\boldsymbol{0}^D)^T)
\end{pmatrix}
$$

Layer 2:

$$
\underbrace{\begin{pmatrix}
\boldsymbol{x}_1 & \dots & \boldsymbol{x}_N & \mathbf{0}^D \\
\boldsymbol{y}_1 & \dots & \boldsymbol{y}_N & 0 \\
\mathbf{0}^D & \dots & \mathbf{0}^D & \boldsymbol{w}_0 \\
\multicolumn{4}{c}{\leftarrow \sum_{i=1}^N \boldsymbol{y}_i \boldsymbol{x}_i \rightarrow} \\
\multicolumn{4}{c}{\leftarrow \sum_{i=1}^N flt(\boldsymbol{x}_i \boldsymbol{x}_i^T) \rightarrow}
\end{pmatrix}}_{conv(in\_proj(\cdot))}
\odot
\underbrace{\begin{pmatrix}
\leftarrow \mathbf{1}^D \rightarrow \\
\leftarrow 1 \rightarrow \\
\leftarrow \mathbf{1}^D \rightarrow \\
\leftarrow \mathbf{1}^D \rightarrow \\
\leftarrow \mathbf{1}^{D^2} \rightarrow
\end{pmatrix}}_{gate\_proj(\cdot)}
=
\begin{pmatrix}
\boldsymbol{x}_1 & \dots & \boldsymbol{x}_N & \mathbf{0}^D \\
\boldsymbol{y}_1 & \dots & \boldsymbol{y}_N & 0 \\
\mathbf{0}^D & \dots & \mathbf{0}^D & \boldsymbol{w}_0 \\
\multicolumn{4}{c}{\leftarrow \sum_{i=1}^N \boldsymbol{y}_i \boldsymbol{x}_i \rightarrow} \\
\multicolumn{4}{c}{\leftarrow \sum_{i=1}^N flt(\boldsymbol{x}_i \boldsymbol{x}_i^T) \rightarrow}
\end{pmatrix}
$$

$$
\begin{pmatrix}
\boldsymbol{x}_1 & \dots & \boldsymbol{x}_N & \mathbf{0}^D \\
\boldsymbol{y}_1 & \dots & \boldsymbol{y}_N & 0 \\
\mathbf{0}^D & \dots & \mathbf{0}^D & \boldsymbol{w}_0 \\
\multicolumn{4}{c}{\leftarrow \sum_{i=1}^N \boldsymbol{y}_i \boldsymbol{x}_i \rightarrow} \\
\multicolumn{4}{c}{\leftarrow \sum_{i=1}^N flt(\boldsymbol{x}_i \boldsymbol{x}_i^T) \rightarrow}
\end{pmatrix}
\underset{out\_proj=Identity}{\overrightarrow{\phantom{xxxx}}}
\begin{pmatrix}
\boldsymbol{x}_1 & \dots & \boldsymbol{x}_N & \mathbf{0}^D \\
\boldsymbol{y}_1 & \dots & \boldsymbol{y}_N & 0 \\
\mathbf{0}^D & \dots & \mathbf{0}^D & \boldsymbol{w}_0 \\
\multicolumn{4}{c}{\leftarrow \sum_{i=1}^N \boldsymbol{y}_i \boldsymbol{x}_i \rightarrow} \\
\multicolumn{4}{c}{\leftarrow \sum_{i=1}^N flt(\boldsymbol{x}_i \boldsymbol{x}_i^T) \rightarrow}
\end{pmatrix}
$$

Now, we use a single BASECONV layer to implement a gradient descent update.

$$
\underbrace{\begin{pmatrix}
\boldsymbol{x}_1 & \dots & \boldsymbol{x}_N & \mathbf{0}^D \\
\boldsymbol{y}_1 & \dots & \boldsymbol{y}_N & 0 \\
\mathbf{0}^D & \dots & \mathbf{0}^D & \boldsymbol{w}_0 \\
\mathbf{0}^D & \dots & \mathbf{0}^D & \mathbf{1}^D \\
\multicolumn{4}{c}{\leftarrow \sum_{i=1}^N \boldsymbol{y}_i \boldsymbol{x}_i \rightarrow} \\
\multicolumn{4}{c}{\leftarrow \sum_{i=1}^N flt(\boldsymbol{x}_i \boldsymbol{x}_i^T) \rightarrow} \\
\multicolumn{4}{c}{\leftarrow \sum_{i=1}^N \boldsymbol{y}_i \boldsymbol{x}_i \rightarrow} \\
\multicolumn{4}{c}{\leftarrow \sum_{i=1}^N flt(\boldsymbol{x}_i \boldsymbol{x}_i^T) \rightarrow}
\end{pmatrix}}_{conv(in\_proj(\cdot))}
\odot
\underbrace{\begin{pmatrix}
\leftarrow \mathbf{1}^D \rightarrow \\
\leftarrow 1 \rightarrow \\
\leftarrow \mathbf{1}^D \rightarrow \\
\leftarrow \mathbf{1}^D \rightarrow \\
\leftarrow \mathbf{1}^D \rightarrow \\
\leftarrow \mathbf{1}^{D^2} \rightarrow \\
\mathbf{0}^D \ \dots \ \mathbf{0}^D \ \mathbf{1}^D \\
\mathbf{0}^{D^2} \ \dots \ \mathbf{0}^{D^2} \ flt(\mathbf{1}^D \boldsymbol{w}_0^T)
\end{pmatrix}}_{gate\_proj(\cdot)}
=
\begin{pmatrix}
\boldsymbol{x}_1 & \dots & \boldsymbol{x}_N & \mathbf{0}^D \\
\boldsymbol{y}_1 & \dots & \boldsymbol{y}_N & 0 \\
\mathbf{0}^D & \dots & \mathbf{0}^D & \boldsymbol{w}_0 \\
\mathbf{0}^D & \dots & \mathbf{0}^D & \mathbf{1}^D \\
\multicolumn{4}{c}{\leftarrow \sum_{i=1}^N \boldsymbol{y}_i \boldsymbol{x}_i \rightarrow} \\
\multicolumn{4}{c}{\leftarrow \sum_{i=1}^N flt(\boldsymbol{x}_i \boldsymbol{x}_i^T) \rightarrow} \\
\mathbf{0}^D \ \dots \ \mathbf{0}^D \ \sum_{i=1}^N \boldsymbol{y}_i \boldsymbol{x}_i \\
\mathbf{0}^{D^2} \ \dots \ \mathbf{0}^{D^2} \ \sum_{i=1}^N flt(\boldsymbol{x}_i (\boldsymbol{x}_i \odot \boldsymbol{w}_0)^T)
\end{pmatrix}
$$

Note that the gradient

$$
\nabla_{\boldsymbol{w}} \mathcal{L}(\boldsymbol{w}_0) = \frac{1}{N}\left( \sum_{i=1}^N \boldsymbol{y}_i \boldsymbol{x}_i - \left( \sum_{i=1}^N \boldsymbol{x}_i \boldsymbol{x}_i^T \right) \boldsymbol{w_0} \right)
$$

can be written as a linear combination of the vector

$$
\begin{pmatrix}
\sum_{i=1}^N \boldsymbol{y}_i \boldsymbol{x}_i \\
\sum_{i=1}^N flt(\boldsymbol{x}_i (\boldsymbol{x}_i \odot \boldsymbol{w}_0)^T)
\end{pmatrix}
$$

so we can write a weight construction for $out\_proj$ that updates $w_0 \to w_0 - \eta \nabla_{\boldsymbol{w}} \mathcal{L}(\boldsymbol{w}_0)$:

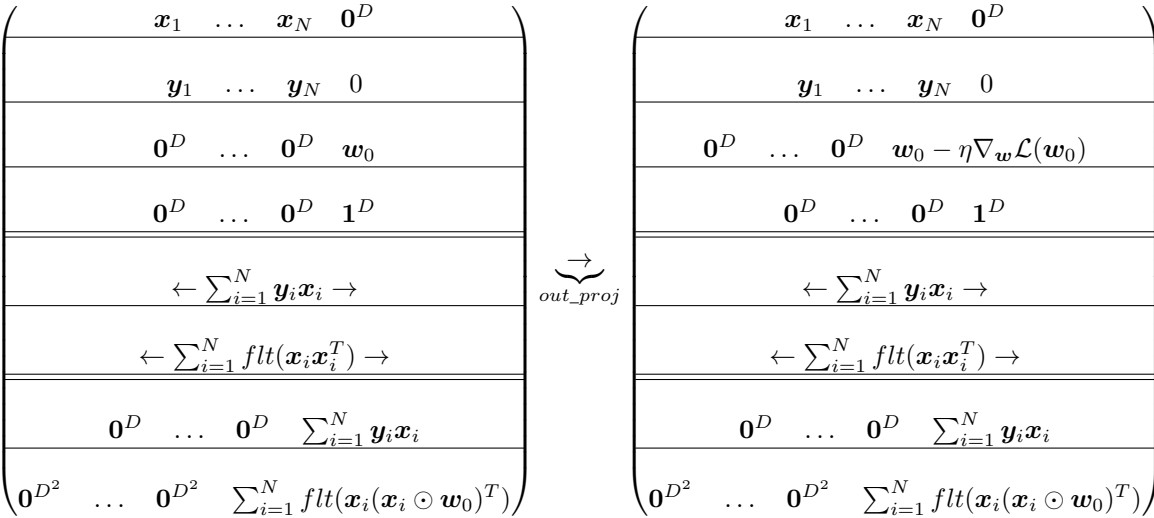

### D.3.2. LOWER BOUNDS: BASECONV CONSTRUCTIONS ARE ASYMPTOTICALLY OPTIMAL

Note that the non-causal weight construction in Appendix D.3.1 requires $O(1)$ layers and $O(D)$ state size, while the causal weight construction in Appendix D.3.1 requires $O(1)$ layers and $O(D^2)$ state size. Clearly the $O(D)$ state size requirement for non-causal models is tight, since one needs to store the gradient $\nabla_{\boldsymbol{w}} \mathcal{L} \in \mathbb{R}^D$. In this section, we prove that the $O(D^2)$ state size requirement for causal models is also asymptotically tight.

**Theorem D.32.** *Any single-pass (causal) algorithm computing the gradient*

$$\nabla_{\boldsymbol{w}} \mathcal{L} = \frac{1}{N} \left( \sum_{j=1}^{N} y_j \boldsymbol{x}_j - \left( \sum_{j=1}^{N} \boldsymbol{x}_j \boldsymbol{x}_j^T \right) \boldsymbol{w} \right)$$

*given inputs $\{(\boldsymbol{x}_1, y_1), \ldots, (\boldsymbol{x}_N, y_N); \boldsymbol{w}\}$, with $(\boldsymbol{x}_i, y_i) \in \mathbb{R}^{(D+1)N}$ and $\boldsymbol{w} \in \mathbb{R}^D$, requires $\Omega(D^2)$ state size in the worst case, where $y_j \in \mathbb{R}$ and $\boldsymbol{x}_j, \boldsymbol{w} \in \mathbb{R}^D$.*

*Proof.* For simplicity, we pick $N = D$ for large enough $D$.

Since we can compute $\frac{1}{N} \sum_{j=1}^{D} y_j \boldsymbol{x}_j$ in $O(D)$ space, we focus on computing the expensive $\left( \sum_{j=1}^{N} \boldsymbol{x}_j \boldsymbol{x}_j^T \right) \boldsymbol{w}$ term. Assume there exists a single-pass algorithm $\mathcal{A}$ that computes $\left( \sum_{j=1}^{N} \boldsymbol{x}_j \boldsymbol{x}_j^T \right) \boldsymbol{w}$ exactly for all choices of $\boldsymbol{x}_1, \ldots, \boldsymbol{x}_D, \boldsymbol{w} \in \mathbb{R}^D$. Now consider the following two claims:

1. Define $\boldsymbol{s}_D$ to be the state of the algorithm after seeing $\boldsymbol{x}_1, \ldots, \boldsymbol{x}_D$. Then we claim that $\boldsymbol{s}_D$ must have enough information to exactly reconstruct $\boldsymbol{M}_D := \sum_{j=1}^{D} \boldsymbol{x}_j \boldsymbol{x}_j^T$.

   This follows since the algorithm must be correct for any value $\boldsymbol{w} \in \mathbb{R}^D$ takes on. In particular, setting $\boldsymbol{w} = \boldsymbol{e}_i$ for $i \in [D]$, we observe that the algorithm must be able to exactly recover $\boldsymbol{M}_D \boldsymbol{e}_i = \boldsymbol{M}_D[:, i]$, $i \in [D]$.

2. The space of matrices

$$\left\{ \sum_{j=1}^{D} \boldsymbol{x}_j \boldsymbol{x}_j^T \right\}$$

   over all choices of $\boldsymbol{x}_j \in \mathbb{R}^D$, $j \in [d]$ contains the set of all real symmetric matrices in $\mathbb{R}^{D \times D}$.

   This holds since for any real symmetric matrix $\boldsymbol{A}$, we can obtain a set of possible $\boldsymbol{x}_j$'s via its eigendecomposition (Strang, 2012):

$$\boldsymbol{A} = \boldsymbol{Q} \boldsymbol{\Lambda} \boldsymbol{Q}^T = \sum_{j=1}^{D} \boldsymbol{x}_j \boldsymbol{x}_j^T$$

where $x_j = \sqrt{\lambda_j} Q[:,j]$.

From the first claim, we conclude that $s_D$ must contain enough information to be able to recover $M_D$ for any possible value $M_D$ can take on (over all choices of $x_1, \ldots, x_D \in \mathbb{R}^D$). From the second claim, we have that the space of possible values of $M_D$ includes the set of all possible real symmetric matrices. Since we know that this set requires $\frac{(D)(D+1)}{2}$ parameters to represent, we can conclude that $|s_D| \geq \frac{(D)(D+1)}{2} \geq \Omega(D^2)$. $\qquad\square$

## D.4. BASECONV and Jackson's Theorem

In this section we prove BASECONV's ability to approximate arbitrary univariate and multivariate smooth functions.

We start with a special case of smooth functions that apply entry-wise univariate smooth functions:

**Definition D.33.** Let $\overline{f} : [-1, 1] \to \mathbb{R}$ be a $(k, L)$-smooth univariate function. Then define

$$f : [-1, 1]^{N \times D} \to \mathbb{R}^{N \times D}$$

as follows. For all $0 \le i < N, 0 \le j < D$, and $\mathbf{u} \in [-1, 1]^{N \times D}$:

$$(f(\mathbf{u}))[i, j] = \overline{f}(\mathbf{u}[i, j]).$$

Now we will state a simple observation on BASECONV's ability to approximate these functions.

**Lemma D.34.** *For any smooth function $f$ as defined in Definition D.33, let $g(\mathbf{x}) = P_{\overline{f}}(\mathbf{x})$ with $P_{\overline{f}}$ being the polynomial from Corollary D.12. Then for all $\mathbf{x} \in [-1, 1]^{N \times D}$,*

$$\|g(\mathbf{x}) - f(\mathbf{x})\|_\infty \le \epsilon.$$

*Proof.* Follows from Definitions D.7 and D.33 and Corollary D.12. □

Next we will state a construction of an arithmetic circuit for a function that applies a univariate polynomial to all entries in $[-1, 1]^{N \times D}$:

**Lemma D.35.** *Let $P(X)$ be a degree $d$ univariate polynomial. Then there is a $(ND, O(ND), O(d), ND)$-circuit to compute $P(\mathbf{u})$ where $P(\mathbf{u})$ is defined as follows. For an input $\mathbf{u} \in [-1, 1]^{N \times D}$,*

$$P(\mathbf{u})[i, j] = P(\mathbf{u}[i, j]).$$

*Proof.* Let the univariate polynomial be

$$P(X) = \sum_{i=0}^{d} c_i X^i$$

where coefficients $c_i \in \mathbb{R}$.

Next we state the natural arithmetic circuit to compute $P(x)$ for $x \in \mathbb{R}$ in Algorithm 2:

---

**Algorithm 2** circuit $\mathcal{C}_P(x)$:

---
1: $s_0 \leftarrow c_0$
2: $m_0 \leftarrow 1$
3: **for** $j = 1, 2, \ldots, d$ **do**
4:     $m_j \leftarrow m_{j-1} \cdot x$         ▷ Multiplication gate
5:     $t_j \leftarrow c_j \cdot m_j$         ▷ Multiplication gate
6:     $s_j \leftarrow s_{j-1} + t_j$         ▷ Addition gate
7: **return** $s_d$         ▷ $s_d$ is the output gate

---

Next we apply the above circuit in parallel to form the circuit that computes $P(\mathbf{u})$ in Algorithm 3:

---

**Algorithm 3** Circuit for $P(\mathbf{u})$:

---
1: **for** $i = 0, 1, \ldots, N - 1$ **do**
2:     **for** $j = 0, 1, \ldots, D - 1$ **do**
3:         $\mathbf{z}[i, j] = \mathcal{C}_P(\mathbf{u}[i, j])$         ▷ Do this in parallel
4: **return** $\mathbf{z}$         ▷ $\mathbf{z}$ is the output matrix

---

Looking at Algorithm 2, the depth of the circuit is $3d$, or $O(d)$, since that is the bound on iterations of the for loop, and each iteration we compute 3 sequential operations. Therefore it's a $(1, O(d), O(d, O(1))$-*circuit*.

For Algorithm 3, The width is $O(ND)$, since we have our input of size $N \times D$, which goes through the circuit in parallel, as stated in Algorithm 3. Therefore we have an $(ND, O(ND), O(d), O(ND))$-*circuit* that computes $P(\mathbf{u})$. □

Since BASECONV has the ability to represent any arithmetic circuit, we get the following:

**Corollary D.36.** *We can implement $P(\mathbf{u})$ (where $P(\mathbf{u})$ is as defined in Lemma D.35 ) when $\deg(P) = d$ with a $(N, O(d \log(ND)), D, O(ND), D) - $ BASECONV.*

*Proof.* Follows from Lemma D.35 giving us the $(ND, O(ND), O(d), O(ND))$-*circuit* for an arbitrary polynomial and Theorem D.30 gives us the BASECONV model to implement the circuit. □

We will prove a tighter bound showing we can represent $P(\mathbf{u})$ using a constant number of BASECONV layers (for constant $\deg(P)$):

**Theorem D.37.** *We can implement $P(\mathbf{u})$ when $\deg(P) = d$ with an $(O(N), O(d), D, O(N), D) - $ BASECONV model.*

*Proof.* We will convert the steps done in Algorithm 2 to layers of BASECONV. Since Algorithm 3 is essentially running Algorithm 2 in parallel over all entries of input $\mathbf{u} \in [-1, 1]^{N \times D}$, the latter happens automatically in our BASECONV implementation.

For this proof, define

$$P_j(X) = X^j$$

and let $\mathbf{C}_i$ be the matrix of size $N \times D$ and all the entries are $c_i$.

We expand the input to our BASECONV layers as follows,

$$\mathbf{u} = \begin{pmatrix} \mathbf{u}' \\ \mathbf{0}^{3N \times D} \end{pmatrix}.$$

This means that the size of the internal dimension of our BASECONV layers will be $(4N, D)$.

To begin iterations of the for loop we need to store initial values into the extra space in $\mathbf{u}$. Taking us from

$$\mathbf{u} = \begin{pmatrix} \mathbf{u}' \\ \mathbf{0}^{N \times D} \\ \mathbf{0}^{N \times D} \\ \mathbf{0}^{N \times D} \end{pmatrix} \rightarrow \begin{pmatrix} \mathbf{u} \\ \mathbf{1}^{N \times D} \\ \mathbf{1}^{N \times D} \\ \mathbf{C}_0 \end{pmatrix} =: \mathbf{u}_0$$

We do this via $\text{BASECONV}(\mathbf{u}', \mathbf{I}^{D \times D}, \begin{pmatrix} \mathbf{0}^{N \times D} \\ \mathbf{1}^{N \times D} \\ \mathbf{1}^{N \times D} \\ \mathbf{C}_0 \end{pmatrix}, \mathbf{0}^{4N \times D}, \mathbf{1}^{4N \times D})$ which computes

$$\left( \left( \begin{pmatrix} \mathbf{u} \\ \mathbf{0}^{N \times D} \\ \mathbf{0}^{N \times D} \\ \mathbf{0}^{N \times D} \end{pmatrix} \mathbf{I}^{D \times D} + \begin{pmatrix} \mathbf{0}^{N \times D} \\ \mathbf{1}^{N \times D} \\ \mathbf{1}^{N \times D} \\ \mathbf{C}_0 \end{pmatrix} \right) \odot \left( \mathbf{0}^{4N \times D} * \begin{pmatrix} \mathbf{u} \\ \mathbf{0}^{N \times D} \\ \mathbf{0}^{N \times D} \\ \mathbf{0}^{N \times D} \end{pmatrix} + \mathbf{1}^{4N \times D} \right) \right).$$

The above simplifies to

$$\left( \left( \begin{pmatrix} \mathbf{u} \\ \mathbf{0}^{N \times D} \\ \mathbf{0}^{N \times D} \\ \mathbf{0}^{N \times D} \end{pmatrix} + \begin{pmatrix} \mathbf{0}^{N \times D} \\ \mathbf{1}^{N \times D} \\ \mathbf{1}^{N \times D} \\ \mathbf{C}_0 \end{pmatrix} \right) \odot \left( \mathbf{1}^{4N \times D} \right) \right),$$

which gives us

$$\begin{pmatrix} \mathbf{u} \\ \mathbf{1}^{N \times D} \\ \mathbf{1}^{N \times D} \\ C_0 \end{pmatrix} =: \mathbf{u}_0,$$

as desired

This was done with a $(4N, 1, D, 4N, D) - \textsc{BaseConv}$ layer.

Our goal is, at the end of iteration $j$ to compute $\mathbf{u}_j \in \mathbb{R}^{4N \times D}$ such that,

$$\mathbf{u}_j = \begin{pmatrix} \mathbf{u} \\ \hline P_j(\mathbf{u}) \\ \hline C_j \odot P_j(\mathbf{u}) \\ \hline C_0 + C_1 \odot P_1(\mathbf{u}) + \cdots + C_j \odot P_j(\mathbf{u}) \end{pmatrix}.$$

We will view the above matrix in terms of the variables in the Algorithm 2 as follows

$$\begin{pmatrix} \mathbf{u} \\ \hline P_j(\mathbf{u}) \\ \hline C_j \odot P_j(\mathbf{u}) \\ \hline C_0 + C_1 \odot P_1(\mathbf{u}) + \cdots + C_j \odot \mathbf{u}^j \end{pmatrix} =: \begin{pmatrix} \mathbf{u} \\ \mathbf{m}_j \\ \mathbf{t}_j \\ \mathbf{s}_j \end{pmatrix}.$$

The for loop runs for values of $1 \le j \le d$ which the remainder of this proof will replicate. There are three lines in the for loop in Algorithm 2 which we will cover how these operations happen in constant number of $\textsc{BaseConv}$ layers.

In line 4, the first line in the for loop computes

$$\mathbf{u}_{j-1} = \begin{pmatrix} \mathbf{u} \\ \mathbf{m}_{j-1} \\ \mathbf{t}_{j-1} \\ \mathbf{s}_{j-1} \end{pmatrix} \rightarrow \begin{pmatrix} \mathbf{u} \\ \mathbf{m}_j \\ \mathbf{t}_{j-1} \\ \mathbf{s}_{j-1} \end{pmatrix} =: \mathbf{u}_j^{(1)}.$$

Note that $\mathbf{m}_j = \mathbf{m}_{j-1} \odot \mathbf{u}$.

We use the `remember` primitive to compute $\mathbf{u}_j^{(1)}$ from $\mathbf{u}_{j-1}$. Define $f : \mathbb{R}^{2N \times D} \to \mathbb{R}^{2N \times D}$ as follows

$$f \begin{pmatrix} \mathbf{u} \\ \mathbf{m}_{j-1} \end{pmatrix} = \begin{pmatrix} \mathbf{u} \\ \mathbf{m}_{j-1} \odot \mathbf{u} \end{pmatrix}.$$

If we can compute $f$ with $\textsc{BaseConv}$ layers then we can compute $\mathbf{u}_j^{(1)}$ for $\mathbf{u}_{j-1}$ by calling `remember`$(\mathbf{u}_j, 0, 2N - 1, f)$.

We show $\textsc{BaseConv} \left( \begin{pmatrix} \mathbf{u} \\ \mathbf{m}_j \end{pmatrix}, \mathbf{I}^{D \times D}, \mathbf{0}^{2N \times D}, \mathbf{H}, \begin{pmatrix} \mathbf{1}^{N \times D} \\ \mathbf{0}^{N \times D} \end{pmatrix} \right)$ maps

$$\begin{pmatrix} \mathbf{u} \\ \mathbf{m}_{j-1} \end{pmatrix} \rightarrow \begin{pmatrix} \mathbf{u} \\ \mathbf{m}_j \end{pmatrix},$$

where $\mathbf{H}$ is defined as in Proposition D.24. We plug the matrices into the $\textsc{BaseConv}$ layer as follows:

$$\left( \begin{pmatrix} \mathbf{u} \\ \mathbf{m}_{j-1} \end{pmatrix} \cdot \mathbf{I}^{D \times D} + \mathbf{0}^{2N \times D} \right) \odot \left( \mathbf{H} * \begin{pmatrix} \mathbf{u} \\ \mathbf{m}_{j-1} \end{pmatrix} + \begin{pmatrix} \mathbf{1}^{N \times D} \\ \mathbf{0}^{N \times D} \end{pmatrix} \right).$$

We know from Proposition D.24 that this convolution operation is a shift down by $N$ rows. Therefore the above simplifies to

$$\left( \begin{pmatrix} \mathbf{u} \\ \mathbf{m}_{j-1} \end{pmatrix} \cdot \mathbf{I}^{D \times D} + \mathbf{0}^{2N \times D} \right) \odot \left( \begin{pmatrix} \mathbf{0}^{N \times D} \\ \mathbf{u} \end{pmatrix} + \begin{pmatrix} \mathbf{1}^{N \times D} \\ \mathbf{0}^{N \times D} \end{pmatrix} \right),$$

which simplifies to

$$\begin{pmatrix} \mathbf{u} \\ \mathbf{m}_{j-1} \end{pmatrix} \odot \begin{pmatrix} \mathbf{1}^{N \times D} \\ \mathbf{u} \end{pmatrix} = \begin{pmatrix} \mathbf{u} \\ \mathbf{m}_{j-1} \odot \mathbf{u} \end{pmatrix} = f \begin{pmatrix} \mathbf{u} \\ \mathbf{m}_j \end{pmatrix},$$

as desired. Therefore by Proposition D.26, line 4 can be computed by $(4N, 8, D, 4N, D) - \text{BASECONV}$.

For line 5 of the for loop we need to compute

$$\mathbf{u}_j^{(1)} = \begin{pmatrix} \mathbf{u} \\ \mathbf{m}_j \\ \mathbf{t}_{j-1} \\ \mathbf{s}_{j-1} \end{pmatrix} \rightarrow \begin{pmatrix} \mathbf{u} \\ \mathbf{m}_j \\ \mathbf{t}_j \\ \mathbf{s}_{j-1} \end{pmatrix} =: \mathbf{u}_j^{(2)}.$$

Note that $\mathbf{t}_j = \boldsymbol{C}_j \odot \mathbf{m}_j$.

To do this we will use three BASECONV layers. We use the `remember` primitive to compute $\mathbf{u}_j^{(2)}$ from $\mathbf{u}_j^{(1)}$. Define $g : \mathbb{R}^{2N \times D} \rightarrow \mathbb{R}^{2N \times D}$ as follows,

$$g \begin{pmatrix} \mathbf{m}_j \\ \mathbf{t}_{j-1} \end{pmatrix} = \begin{pmatrix} \mathbf{m}_j \\ \boldsymbol{C}_j \odot \mathbf{m}_j \end{pmatrix}.$$

If we can compute $g$ with BASECONV layers then we can compute $\mathbf{u}_j^{(2)}$ for $\mathbf{u}_{j-1}$ by calling $\text{remember}(\mathbf{u}_j^{(1)}, N, 3N-1, g)$.

Indeed, we show the $g$ can be computed by first computing $\text{BASECONV} \left( \begin{pmatrix} \mathbf{m}_j \\ \mathbf{t}_{j-1} \end{pmatrix}, \boldsymbol{I}^{D \times D}, \mathbf{0}^{2N \times D}, \mathbf{0}^{2N \times D}, \begin{pmatrix} \mathbf{1}^{N \times D} \\ \mathbf{0}^{N \times D} \end{pmatrix} \right)$:

$$\left( \begin{pmatrix} \mathbf{m}_j \\ \mathbf{t}_{j-1} \end{pmatrix} \cdot \boldsymbol{I}^{D \times D} + \mathbf{0}^{2N \times D} \right) \odot \left( \mathbf{0}^{2N \times D} * \begin{pmatrix} \mathbf{m}_j \\ \mathbf{t}_{j-1} \end{pmatrix} + \begin{pmatrix} \mathbf{1}^{N \times D} \\ \mathbf{0}^{N \times D} \end{pmatrix} \right),$$

which simplifies to

$$\left( \begin{pmatrix} \mathbf{m}_j \\ \mathbf{t}_{j-1} \end{pmatrix} \right) \odot \left( \begin{pmatrix} \mathbf{1}^{N \times D} \\ \mathbf{0}^{N \times D} \end{pmatrix} \right).$$

This results in

$$\begin{pmatrix} \mathbf{m}_j \\ \mathbf{0}^{N \times D} \end{pmatrix}.$$

We pass into the next layer, $\text{BASECONV} \left( \begin{pmatrix} \mathbf{m}_j \\ \mathbf{0}^{N \times D} \end{pmatrix}, \boldsymbol{I}^{D \times D}, \begin{pmatrix} \mathbf{0}^{N \times D} \\ \mathbf{1}^{N \times D} \end{pmatrix}, \boldsymbol{H}, \begin{pmatrix} \mathbf{1}^{N \times D} \\ \mathbf{0}^{N \times D} \end{pmatrix} \right)$ where $\boldsymbol{H}$ is defined as in Proposition D.24:

$$\left( \begin{pmatrix} \mathbf{m}_j \\ \mathbf{0}^{N \times D} \end{pmatrix} \cdot \boldsymbol{I}^{D \times D} + \begin{pmatrix} \mathbf{0}^{N \times D} \\ \mathbf{1}^{N \times D} \end{pmatrix} \right) \odot \left( \boldsymbol{H} * \begin{pmatrix} \mathbf{m}_j \\ \mathbf{0}^{N \times D} \end{pmatrix} + \begin{pmatrix} \mathbf{1}^{N \times D} \\ \mathbf{0}^{N \times D} \end{pmatrix} \right).$$

Since the kernel $\boldsymbol{H}$ is as in Proposition D.24, this simplifies to

$$\left( \begin{pmatrix} \mathbf{m}_j \\ \mathbf{1}^{N \times D} \end{pmatrix} \odot \left( \begin{pmatrix} \mathbf{0}^{N \times D} \\ \mathbf{m}_j \end{pmatrix} + \begin{pmatrix} \mathbf{1}^{N \times D} \\ \mathbf{0}^{N \times D} \end{pmatrix} \right) \right).$$

The above simplifies further to

$$\begin{pmatrix} \mathbf{m}_j \\ \mathbf{1}^{N \times D} \end{pmatrix} \odot \begin{pmatrix} \mathbf{1}^{N \times D} \\ \mathbf{m}_j \end{pmatrix},$$

which results in:

$$\begin{pmatrix} \mathbf{m}_j \\ \mathbf{m}_j \end{pmatrix}.$$

We pass the above to $\text{BASECONV} \left( \begin{pmatrix} \mathbf{m}_j \\ \mathbf{m}_j \end{pmatrix}, \boldsymbol{I}^{D \times D}, \mathbf{0}^{2N \times D}, \mathbf{0}^{2N \times D}, \begin{pmatrix} \mathbf{1}^{N \times D} \\ \boldsymbol{C}_j \end{pmatrix} \right)$:

$$\left( \begin{pmatrix} \mathbf{m}_j \\ \mathbf{m}_j \end{pmatrix} \cdot \boldsymbol{I}^{D \times D} + \mathbf{0}^{2N \times D} \right) \odot \left( \mathbf{0}^{2N \times D} * \begin{pmatrix} \mathbf{m}_j \\ \mathbf{m}_j \end{pmatrix} + \begin{pmatrix} \mathbf{1}^{N \times D} \\ \boldsymbol{C}_j \end{pmatrix} \right)$$

which simplifies to

$$\begin{pmatrix} \mathbf{m}_j \\ \mathbf{m}_j \end{pmatrix} \odot \begin{pmatrix} \mathbf{1}^{N \times D} \\ \mathbf{C}_j \end{pmatrix}.$$

The above results in

$$\begin{pmatrix} \mathbf{m}_j \\ \mathbf{C}_j \odot \mathbf{m}_j \end{pmatrix} = g \begin{pmatrix} \mathbf{m}_j \\ \mathbf{t}_{j-1} \end{pmatrix},$$

as desired.

Therefore by Corollary D.27, line 5 was computed by $(4N, O(1), D, 4N, D) - \text{BASECONV}$.

For line 6, the final line of the for loop, we want

$$\mathbf{u}_j^{(2)} = \begin{pmatrix} \mathbf{u} \\ \mathbf{m}_j \\ \mathbf{t}_j \\ \mathbf{s}_{j-1} \end{pmatrix} \rightarrow \begin{pmatrix} \mathbf{u} \\ \mathbf{m}_j \\ \mathbf{t}_j \\ \mathbf{s}_j \end{pmatrix} =: \mathbf{u}_j.$$

Note that $\mathbf{s}_j = \mathbf{s}_{j-1} + \mathbf{t}_j$

Define function $h : \mathbb{R}^{2N \times D} \rightarrow \mathbb{R}^{2N \times D}$ as follows,

$$h \begin{pmatrix} \mathbf{t}_j \\ \mathbf{s}_{j-1} \end{pmatrix} = \begin{pmatrix} \mathbf{t}_j \\ \mathbf{s}_{j-1} + \mathbf{t}_j \end{pmatrix}.$$

If we can compute $h$ with BASECONV layers then we can compute $\mathbf{u}_j$ for $\mathbf{u}_{j-1}$ by calling $\texttt{remember}(\mathbf{u}_j^{(2)}, 2N, 4N-1, h)$.

Indeed we show that $h$ can be computed by computing $\text{BASECONV}\left( \begin{pmatrix} \mathbf{t}_j \\ \mathbf{s}_{j-1} \end{pmatrix}, \mathbf{0}^{D \times D}, \mathbf{1}^{2N \times D}, \overline{\mathbf{H}}, \mathbf{0}^{2N \times D} \right)$, where kernel $\overline{\mathbf{H}} \in \mathbb{R}^{2N \times D}$ is defined as:

$$\overline{\mathbf{H}}[k, :] \equiv \begin{cases} \mathbf{1}^D & \text{if } k \in \{0, N\} \\ \mathbf{0}^D & \text{otherwise.} \end{cases}$$

.

This layer computes

$$\left( \begin{pmatrix} \mathbf{t}_j \\ \mathbf{s}_{j-1} \end{pmatrix} \cdot \mathbf{0}^{2N \times D} + \mathbf{1}^{2N \times D} \right) \odot \left( \overline{\mathbf{H}} * \begin{pmatrix} \mathbf{t}_j \\ \mathbf{s}_{j-1} \end{pmatrix} + \mathbf{0}^{2N \times D} \right).$$

This simplifies to

$$\left( \mathbf{1}^{2N \times D} \right) \odot \left( \overline{\mathbf{H}} * \begin{pmatrix} \mathbf{t}_j \\ \mathbf{s}_{j-1} \end{pmatrix} \right) = \left( \overline{\mathbf{H}} * \begin{pmatrix} \mathbf{t}_j \\ \mathbf{s}_{j-1} \end{pmatrix} \right).$$

Now we compute this convolution for column $i$, $0 \le i < 2N$. For notational convenience, let $\begin{pmatrix} \mathbf{t}_j \\ \mathbf{s}_{j-1} \end{pmatrix}$ be noted as matrix $\mathbf{V}$. Then we have:

$$\overline{\mathbf{H}}[:, i] * \mathbf{V}[:, i] = \text{coeff} \left( (1 + X^N) \mathbf{V}[:, i](X) \mod X^{2N} \right),$$

where $(1 + X^N)$ is the polynomial representation of the columns of $\overline{\mathbf{H}}$ (since there's a one in the 0th index and a one in the $N$th index of each column).

The expression simplifies to

$$\text{coeff} \, \mathbf{V}[:, i](X) + \mathbf{V}[:, i](X) X^N \mod X^{2N},$$

which can be broken down to

$$\text{coeff} \left( (\mathbf{V}[0][i] + \mathbf{V}[1][i]X + \cdots + \mathbf{V}[2N-1][i]X^{2N-1}) \mod X^{2N} \right)$$
$$+ \text{coeff} \left( (\mathbf{V}[0][i]X^N + \mathbf{V}[1][i]X^{N+1} + \cdots + \mathbf{V}[2N-1][i]X^{3N-1}) \mod X^{2N} \right)$$

with the lower order terms in the second coefficient vector being zeros,

$$\text{coeff}\left(\left(\mathbf{V}[0][i] + \mathbf{V}[1][i]X + \cdots + \mathbf{V}[2N-1][i]X^{2N-1}\right) \mod X^{2N}\right)$$
$$+ \text{coeff}\left(\left(0 + 0X + \cdots + 0X^{N-1} + \mathbf{V}[0][i]X^N + \cdots + \mathbf{V}[2N-1][i]X^{3N-1}\right) \mod X^{2N}\right)$$

After taking $\mod X^{2N}$ we get

$$\text{coeff}\left(\mathbf{V}[0][i] + \mathbf{V}[1][i]X + \cdots + \mathbf{V}[2N-1][i]X^{2N-1}\right)$$
$$+ \text{coeff}\left(0 + 0X + \cdots 0X^{N-1}\mathbf{V}[0][i]X^N + \cdots \mathbf{V}[N-1][i]X^{2N-1}\right)$$

The first set of coefficients is the input matrix as is. And the second one is the input matrix shifted down as seen in Proposition D.24. Therefore when we add these vectors we are doing

$$\begin{pmatrix} \mathbf{t}_j \\ \mathbf{s}_{j-1} \end{pmatrix} + \begin{pmatrix} \mathbf{0}^{N \times D} \\ \mathbf{t}_j \end{pmatrix} = h \begin{pmatrix} \mathbf{t}_j \\ \mathbf{s}_{j-1} \end{pmatrix},$$

as desired. Therefore by Proposition D.26, line 6 is computed with by $(4N, 1, D, 4N, D) - \text{BASECONV}$.

The $\mathbf{s}_d$ matrix gives us $\boldsymbol{C}_0 + \boldsymbol{C}_1 \odot \mathbf{m}_1 + \cdots + \boldsymbol{C}_d \odot \mathbf{m}_d$. Recalling that

$$\boldsymbol{C}_0 + \boldsymbol{C}_1 \odot \mathbf{m}_1 + \cdots + \boldsymbol{C}_d \odot \mathbf{m}_d \equiv \sum_{j=0}^{d} \boldsymbol{C}_j \odot \mathbf{u}^j = P(\mathbf{u}),$$

and hence $\mathbf{s}_d$ is our desired output.

We have $d$ layers, each consisting of $O(1)$ BASECONV layers. Giving us $O(d)$ many layers to implement Algorithm 2.

Therefore, via the ability to stack BASECONV layers to do function composition, the for loop was computed by a $(4N, O(d), D, 4N, D) - \text{BASECONV}$, as desired. □

The following states BASECONV's ability to approximate a univariate smooth function:

**Proposition D.38.** *Let $f$ be the $(k, L)$ -smooth function defined in Definition D.33. Then there is a* $\left(N, O\left(\sqrt[k]{\frac{L}{\epsilon}}\right) + k, D, (ND), D\right) - \text{BASECONV}$ *model that approximates $f$ within error $\epsilon$.*

*Proof.* Follows from Corollary D.12, Lemma D.34, and Theorem D.37. □

### D.5. Multivariate function approximation

We begin by defining more multivariate notation.

We consider the following multivariate functions:

**Definition D.39.** *For $0 \leq 1 < N, 0 \leq j < D$, let $\bar{f}_{i,j} : [-1, 1]^{N \times D} \to \mathbb{R}$ be a $(k, L)$-smooth multivariate function. Then* define

$$f(\mathbf{x}) : [-1, 1]^{N \times D} \to \mathbb{R}^{N \times D}$$

as follows. For all $0 \leq i < N, 0 \leq j < D, \mathbf{u} \in [-1, 1]^{N \times D}$ define

$$f(\mathbf{u})[i, j] := \bar{f}_{i,j}(\mathbf{u}).$$

**Lemma D.40.** *For any smooth function $f$ as defined in Definition D.39, let $g(X_1, \ldots, X_{N \times D}) = P_{\bar{f}}(X_1, \ldots, X_{N \times D})$ be the polynomial from Corollary D.14. Then for all $\mathbf{x} \in [-1, 1]^{N \times D}$,*

$$\|g(\mathbf{x}) - f(\mathbf{x})\|_\infty \leq \epsilon.$$

*Proof.* Follows from Definitions D.7 and D.39 and Corollary D.14. □

Next we will state a construction for an arithmetic circuit for a function that takes a $[-1, 1]^{N \times D}$ variable input:

**Lemma D.41.** *Let $P(\boldsymbol{X})$ be a degree $d$ multivariate polynomial. Then there is a $\big(n, O(d \cdot n^d), O(d \log(n)), O(n^d)\big)$-circuit to compute $P(\mathbf{u})$ on any input $\mathbf{u} \in [-1, 1]^n$.*

*Proof.* Let the multivariate polynomial be as defined in Definition D.6. We build the circuit to compute this in Algorithm 4,

---

**Algorithm 4** circuit $\mathcal{C}_P(\mathbf{x})$:

1: **for** $\boldsymbol{\alpha} = (\alpha_1, \dots, \alpha_n) \in \mathbb{Z}_{\geq 0}^n$ such that $\sum_{i=1}^n \alpha_i \leq d$ **do**
2: $\quad m_{\boldsymbol{\alpha}} \leftarrow 1$
3: $\quad$ **for** $i = 1, 2, \dots, n$ **do** $\qquad\qquad\qquad\qquad\qquad\qquad\qquad\qquad\qquad$ ▷ Done in parallel
4: $\qquad$ **if** $\alpha_i \neq 0$ **then**
5: $\qquad\quad m_{\boldsymbol{\alpha}} \leftarrow m_{\boldsymbol{\alpha}} \cdot x_i^{\alpha_i}$
6: $\quad t_{\boldsymbol{\alpha}} \leftarrow c_{\boldsymbol{\alpha}} \cdot m_{\boldsymbol{\alpha}}$
7: **for** $\boldsymbol{\alpha} = (\alpha_1, \dots, \alpha_n) \in \mathbb{Z}_{\geq 0}^n$ such that $\sum_{i=1}^n \alpha_i \leq d$ **do**
8: $\quad s \leftarrow \sum t_{\boldsymbol{\alpha}}$ $\qquad\qquad\qquad\qquad\qquad\qquad\qquad\qquad\qquad\qquad\qquad$ ▷ Done in parallel
9: **return** s

---

We compute the for loop starting on line 3 by making multiplications in parallel. Therefore obtaining a depth of $O(\log(d))$. We also have the for loop starting on line 7, making pairwise addition operations, resulting in a depth of $O(d \log(n))$. □

We again use the result that BASECONV can represent any arithmetic circuit to get:

**Corollary D.42.** *We can implement $P(\mathbf{u})$ (where $P(\mathbf{u})$ is as defined in Lemma D.41) when $\deg(P(X_1, \dots, X_{ND})) = d$ with a $\big(N, O(d \log(ND)), D, O((ND)^d), D\big) - $ BASECONV where $\mathbf{u} \in [-1, 1]^{N \times D}$.*

*Proof.* Lemma D.41 gives us the arithetmic circuit that computes this polynomial. Then via Theorem D.30 we get a $\big(N, O(d \log(ND)), D, O((ND)^d), D\big) - $ BASECONV model to implement the circuit. □

Finally we state BASECONV's ability to approximate multivariate smooth functions:

**Proposition D.43.** *Let $f$ be the function defined in Definition D.39. Then there is a $\big(N, O(d \log(ND)), D, O((ND)^d), D\big) - $ BASECONV model that approximates $f$ to within error $\epsilon$, with $d = O_k\big(\sqrt[k]{\frac{NDL}{\epsilon}}\big)$.*

*Proof.* We get the existence of a polynomial that approximates $f$ for some $\epsilon$ from Corollary D.14. Then via Corollary D.42 we get that we can represent any polynomial, implying $\big(N, O(d \log(ND)), D, O((ND)^d), D\big) - $ BASECONV represents any polynomial that approximates the multivariate smooth function $f$. □

