# OpenReview forum: "Can Transformers Solve Least Squares to High Precision?"
_ICML.cc/2024/Workshop/ICL — ICML 2024 Workshop ICL Poster_

### Official Review · Reviewer_mwtV · 2024-06-07
**Understanding the Error Floor in Using Sequential Model to Solve Linear Regression Problem.**

**Rating:** 2
**Fit:** 3
**Confidence:** 3

**Workshop Review:**

I appreciate the authors' identification of the precision issue in training transformers to learn regression tasks and their extensive experimental and theoretical analysis to investigate the error source. Here are a few questions I had while reading the paper.

The authors have identified the error source in the trained transformer. However, it is still unclear to me why they chose to provide weight construction for BaseConv to implement the linear regression optimizer. There are already papers that construct a transformer model to solve linear regression precisely, such as [Giannou et al. 2023]. It would be great if the authors could highlight the difference between the previous construction and the construction conducted in this paper, as well as the motivations for looking at BaseConv when the goal is to study the error floor for transformers, and the error source results are for transformers. If the goal is to study in general any sequential models' error precision, then other results should probably be included such as Mamba.

In Figure 5, if BaseConv can be constructed to solve linear regression, why is there a mismatch in the curve compared to GD?

Can the insights observed in investigating BaseConv (section 5) be transferred to why transformers have a higher error rate? For example, if we take the construction for transformers and evaluate their performance in a similar way as presented in Figure 7 (right), will transformers with construction in the first few layers close the error gap or have similar performance as partially frozen BaseConv?

When comparing the performance of attention with BaseConv (Figure 7), it is unclear to me whether the performance gap comes from the different configurations of attention vs. BaseConv. Could you report the number of parameters and hyperparameter settings utilized for each method (especially for BaseConv)? And what is the training loss for transformers vs. BaseConv on this gradient learning task?

----
[Giannou et al. 2023]: Giannou, A., Rajput, S., Sohn, J.-Y., Lee, K., Lee, J. D., and Papailiopoulos, D. Looped transformers as programmable computers. In International Conference on Machine Learning, pp. 11398–11442. PMLR, 2023.

**Reason For Not Giving Higher Score:**

It is still unclear to me what the motivation is for studying BaseConv's error precision performance. Most of my questions revolve around this fundamental issue regarding the motivation and goal of this paper. In particular, if the experimental results support that BaseConv is a better alternative than a transformer in preserving precision, then this argument would make more sense to me. However, as the experimental results show, only in a few cases does BaseConv perform better than a transformer. Additionally, BaseConv's weights depend on the sequence length n, whereas transformers can handle arbitrarily long sequences (not exceeding the maximum sequence length).

**Reason For Not Giving Lower Score:**

I appreciate the identification of this error precision problem, and the primitive tasks to identify the error source are from the Multiply operation. The author also conducts extensive theory analysis in showing the expressiveness of BaseConv.

---

### Official Review · Reviewer_8HpE · 2024-06-09
**Review for the paper**

**Rating:** 2
**Fit:** 3
**Confidence:** 3

**Workshop Review:**

This paper analyzes questions about the limitations of Transformers and gated convolutions architectures in scientific computing applications requiring high precision.

The paper provides a series of contributions: first, it underlines the limitation of the Transformer architecture to solve least squares problems to high precision. They suggest that the reasoning lies in that Transformers struggle to learn element-wise multiplications precisely. The paper proposes gated convolutional models as an alternative to softmax attention. Finally, the authors propose a simplified training setup where models are supervised to learn the gradient update rule explicitly. This approach leads to a 2-order of magnitude improvement over parameter-matched Transformers trained through standard in-context learning.

While the paper seems very comprehensive with a lot of varied contributions, I question several main assumptions. First, it has been shown that there exist a linear transformers (von Oswald et al, 2023) and ReLU transformers (Bai et al 2023) that _exactly_ equivalent to a gradient descent. Therefore the computation for N layers should be equal to N iteration of GD for any given N. The authors acknowledge that (paragraph l. 206, c.1) Transformers can express GD, but it is the empirical evidence that is lacking. So it looks like it is an optimization problem, rather than theoretical. Another hypothesis is intrinsic properties of softmax. Finally, the issue can also be attributed to the grokking (also observed in e.g. von Oswald et al, 2023) and issues in optimization. I would love to see a more comprehensive analysis of the optimization parameters and methods that the authors have tried in order to confirm the main assumption that Transformers cannot achieve sufficient accuracy.

Another issue that I see is that potentially the authors might have used float32 precision when training transformers. In this case getting error 1e-7 might be expected. float32 has 23 bits of mantissa, so different ways of computing the "same" floating-point result would be expected to differ by 1 part in $2^{23}$, which is roughly 1e-7. This might be an explanation of the fact that the mult operator is not explained well.

**Reason For Not Giving Higher Score:**

It looks like the main assumption of the paper is not corroborated enough. More basic experiments (as suggested above) might be needed to confirm that transformers indeed can't be trained to achieve high accuracy.

**Reason For Not Giving Lower Score:**

The paper does have a lot of interesting claims that are still interesting to the public. The connection to gated convolutional models and training with respect to the gradient are very interesting and clever ideas.

---

### Meta-Review · Area_Chair_tf7T · 2024-06-16

**Recommendation:** 2

**Metareview:**

The paper provides a thorough analysis of the limitations of Transformers and gated convolutional architectures in scientific computing applications requiring high precision. Reviewers raised concerns about the assumptions and experimental setups and they noted that existing models like linear and ReLU transformers can theoretically perform gradient descent, suggesting the issue might be more about optimization rather than theoretical limitations. Additionally, the potential use of float32 precision during training could explain the observed errors. Reviewers also questioned the choice of BaseConv for implementing the linear regression optimizer and requested a clearer comparison with existing methods. They suggested further analysis of optimization parameters and methods to validate the main assumptions. Despite these concerns, the paper's identification of precision issues and extensive experimental and theoretical analysis make it a valuable contribution to the workshop, suggesting acceptance.

---

### Decision · Program_Chairs · 2024-06-17

Accept (Poster)